# DRIPPER: TOKEN-EFFICIENT MAIN HTML EXTRACTION WITH A LIGHTWEIGHT LM

## ABSTRACT

Accurately and efficiently extracting main content from general web pages is of great significance for obtaining training data for large models. Using well-pre-trained decoder-only generative language models offers excellent document comprehension capabilities, thereby effectively enhancing parsing quality. However, it remains constrained by issues such as context window length, inference cost, and format hallucination. We present Dripper, an efficient HTML main content extraction framework powered by lightweight language models, which addresses these challenges through four key innovations: (1) We design a specialized HTML simplification algorithm that reduces input token count to 22% compared to raw HTML while preserving critical structural information; (2) We reformulate main content extraction as a semantic block sequence classification task, significantly reducing inference cost; (3) We introduce a controlled decoding mechanism that strictly constrains the output space through logits processors, effectively eliminating hallucination issues common in small-scale models; (4) We propose Main-WebBench, an evaluation dataset containing over 7,800 web pages with meticulously human-annotated main content extraction labels. Experimental results demonstrate that using only a 0.6B parameter model, Dripper achieves state-of-the-art performance across all evaluation benchmarks and outperforms all baseline methods, attaining an ROUGE-N F1 score of 81.58%(83.13% with fall-back strategy) on our proposed MainWebBench dataset.

## 1 INTRODUCTION

The World Wide Web forms the foundational data repository for modern AI, serving as the primary source for training corpora like C4 (Raffel et al., 2020) and for building the knowledge graphs that power large-scale applications (Wang et al., 2019). The sheer scale of this resource is immense, with web archiving projects like Common Crawl (Common Crawl Foundation) preserving billions of new pages each month. This massive volume presents a fundamental challenge for data utilization: the raw, unstructured HTML must first be converted into high-quality, structured data. Accordingly, the development of robust and accurate content extraction techniques has become a critical prerequisite for a wide range of downstream information processing tasks (Vogels et al., 2018b).

The primary obstacle lies in the failure of traditional extraction methods to handle the web's inherent complexity. While HTML standards provide semantic tags with clear intended uses—such as `<article>` for main content or `<aside>` for sidebars—their adoption in practice is highly inconsistent (Wang et al., 2022), rendering simple tag-based rules unreliable. Similarly, heuristic methods based on statistical properties like text or link density often falter. Even pages built from the same template can exhibit vast statistical variations simply due to differences in their core content, undermining the stability of these metrics. Furthermore, vision-based approaches like diffbot (Diffbot, 2025) are often rendered ineffective in large-scale offline processing scenarios. Web archives like Common Crawl typically store only raw HTML, lacking the corresponding CSS files required to render a page as its developer originally intended. These fundamental challenges explain why established tag-based, heuristic, and vision-based methods struggle to achieve both high accuracy and robust generalization. While the semantic understanding of well-trained decoder-only language models offers a promising theoretical solution (Wang et al., 2025), their direct application is thwarted by a distinct set of severe practical barriers. First, **excessive context length** makes processing raw HTML infeasible at scale.Our analysis of 14,000 Common Crawl files shows 29.3% of

pages exceeded 32k tokens and 21.0% surpassed 128k tokens, lengths that far exceed the context windows of most SLMs. Second, the **structural complexity** of HTML presents a critical trade-off. While stripping all tags is an effective way to significantly reduce input length, this action simultaneously destroys the vital structural information they contain. Without these cues, an algorithm cannot reliably distinguish main content from noise and perform accurate extraction. Finally, LLMs are prone to **output hallucination** (Ji et al., 2023), a tendency to generate content not present in the source document, which constitutes a critical failure for an extraction task that demands high fidelity.

To address these challenges, we introduce **Dripper**, a novel framework that reframes web content extraction as an efficient Sequential Block Classification task, specifically designed for Small Language Models (SLMs). Our three-stage pipeline begins with a pre-processing step that simplifies the raw HTML, making it tractable for a compact model. We then employ a 0.6B parameter SLM, **Dripper-0.6B**, to perform a localized binary classification on each semantic block of the simplified document. To ensure perfect output fidelity and eliminate hallucinations, we guide the SLM's decoding with a custom logits processor, forcing it to produce a structured sequence of labels. Finally, a post-processing step uses these high-confidence labels to precisely extract the corresponding content blocks from the original HTML structure. The text from these selected blocks is then evaluated against the ground truth using ROUGE-N F1 as the primary metric. This approach circumvents the context length and hallucination issues inherent in holistic generative methods.

Our main contributions are summarized as follows:

(1) We introduce **a novel HTML simplification algorithm** that strips redundant information while preserving critical structural markers, compressing the average document size by 22% and making processing feasible for SLMs.

(2) The HTML document is represented as **a sequence of semantic blocks**, which transforms the task into a series of localized binary classifications. This approach dramatically reduces the problem's complexity while retaining essential hierarchical and contextual relationships.

(3) We design a **constrained decoding mechanism** using a custom logits processor. This converts the task from open-ended generation to producing a fixed, structured output, thereby systematically eliminating hallucinations and ensuring high-fidelity results.

(4) To facilitate rigorous and comprehensive evaluation, we construct and will publicly release **MainWebBench**, a new large-scale benchmark with over 7,800 meticulously annotated samples, making it seven times larger than any existing public alternative. Our experiments demonstrate that Dripper, using only a 0.6B parameter model, achieves state-of-the-art performance, outperforming all baselines on MainWebBench with a leading F1 score of **81.58%**, which increases to **83.13%** when augmented with a fallback strategy. Our trained model weights[1],code[2] and the MainWebBench benchmark[3] are publicly available.

## 2 RELATED WORK

Main text extraction aims to extract main content from raw HTML while filtering out boilerplate elements such as navigation and advertisements, a critical technique for building high-quality web corpora. The methods for accomplishing this task have evolved through several distinct paradigms, each addressing the limitations of its predecessor.

**Heuristic and rule-based Methods**. Early approaches predominantly relied on manually engineered heuristics to distinguish main content from boilerplate. These methods operate on the observation that content-rich regions differ structurally from noisy elements, using features like text-to-tag ratios (CETR) (Weninger et al., 2010), visual cues from the rendered page (VIPS) (Cai et al., 2003), or a combination of heuristics such as link and stop-word density (Readability (Mozilla, 2015), jusText (Pomikálek, 2011)). While computationally efficient, these methods are often brittle and require continuous maintenance to adapt to evolving web design patterns.

---

[1]https://huggingface.co/anonymous-s2wrvq/Dripper

[2]https://anonymous.4open.science/r/dripper-1825

[3]https://huggingface.co/datasets/anonymous-s2wrvq/MainWebBench

**Supervised Learning Methods**. To move beyond handcrafted rules, subsequent work approached body text extraction as a supervised machine learning problem. This paradigm shift began with classic methods like Boilerpipe (Kohlschütter et al., 2010), Dragnet (Peters & Lecocq, 2013), which treated the task as a classification problem using manually designed features. The advent of deep learning marked a further evolution from feature engineering to representation learning. (Vogels et al., 2018a; Leonhardt et al., 2020; Zhou et al., 2021). To better leverage the hierarchical structure of HTML, subsequent research introduced Graph Neural Networks (GNNs) (Zhou et al., 2021) and Transformer-based architectures like WebFormer (Endrédy & Novák, 2013), which improved extraction accuracy by capturing complex relationships between nodes. While achieving higher accuracy, these models often require substantial labeled data, and their complex architectures incur significant computational overhead.

**Hybrid Systems and Production Tools**. In parallel with academic advancements, a suite of powerful open-source tools has emerged, often blending multiple techniques for practical application. Trafilatura (Barbaresi, 2021) has become a strong baseline by integrating a sophisticated cascade of rules with established algorithms like jusText (Pomikálek, 2011) and Readability (Mozilla, 2015) as fallbacks. Other tools, such as magic-html (opendatalab, 2024), focus on simplifying complex HTML structures before extraction, often as part of larger document AI ecosystems. More recently, frameworks such as crawl4ai (UncleCode, 2024) have adopted an explicitly hybrid architecture, combining rule-based selectors, traditional machine learning, and Large Language Models (LLMs) to provide versatile solutions for AI data pipelines.

**Generative-Language-based Methods**. Recent months have seen rapid progress in decoder-only large language models. Base models pre-trained on massive, high-quality, and highly-diverse corpora have become the de-facto starting point for most NLP tasks. The most representative work in this line is ReaderLM-v2 (Wang et al., 2025), which frames main-content extraction as an HTML-to-Markdown translation problem. Starting from a 1.5 B-parameter Qwen2.5 checkpoint, the authors first extend the context window to 512 k tokens through continual pre-training, then fine-tune with supervised fine-tuning (SFT) and direct-preference optimization (DPO) to produce clean Markdown. This pipeline reuses the open-source model zoo and inference-acceleration stacks already available in the LLM community. Nevertheless, even the official best-practice implementation [4] still expects the full, un-pruned HTML page as input and generates the complete body text in one pass. This incurs heavy computational overhead and, during long-sequence generation, often produces unwanted artifacts such as repetitions or un-escaped HTML tags. Consequently, the potential of SLMs for extraction remains largely untapped.

## 3 METHODOLOGY

In this section, we detail the methodology of our Dripper framework. We begin in §3.1 with an overview of the system's three-stage architecture. Next, in §3.2, we elaborate on the core pre-processing and post-processing modules that enable efficient extraction. We then formally define the task as a sequence labeling problem in §3.3. Finally, in §3.4, we introduce our constrained decoding mechanism, which uses a custom logits processor to eliminate hallucinations.

### 3.1 SYSTEM ARCHITECTURE OVERVIEW

The Dripper framework operates through a three-stage pipeline: pre-processing, SLM-based extraction, and post-processing. As illustrated in Figure1, the system takes a raw HTML document as input and transforms it into a clean, structured Markdown output.

The process begins with the pre-processing module, which takes a raw HTML document and generates two distinct representations. The first is a `Simplified HTML`, which is simplified and chunked. The second is a `Mapping HTML`, which is only chunked but otherwise unmodified. This parallel representation is crucial for ensuring the final extracted content remains a valid subtree of the original Document Object Model (DOM). The `Simplified HTML` is then passed to Dripper-0.6B, which identifies and labels the main content blocks. The decoding process is constrained by a custom logits processor to guarantee the structural integrity and correctness of the output format. Finally, in the post-processing stage, the Dripper-0.6B's classification output is used to prune

---

[4]https://huggingface.co/jinaai/ReaderLM-v2

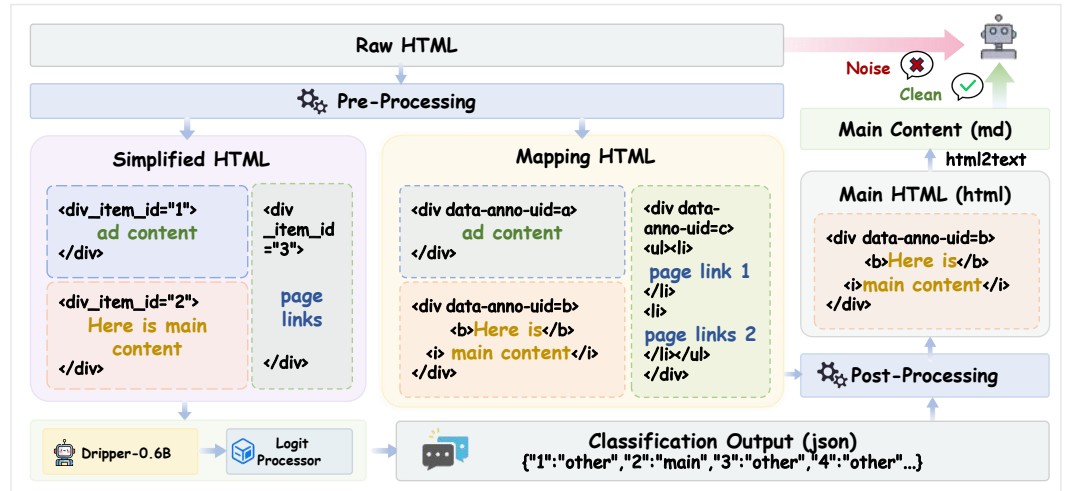

Figure 1: An overview of the Dripper framework, which operates as a three-stage pipeline. (1) Pre-processing: A raw HTML document is converted into two parallel representations: **Simplified HTML** for model input and **Mapping HTML** for final reconstruction. (2) Dripper-0.6B Extraction: Dripper-0.6B performs sequential block classification on the simplified input, guided by a custom logits processor to output a structured sequence. (3) Post-processing: The labels are used to select the corresponding blocks from **Mapping HTML** to construct the final, clean **Main Content**.

the `Mapping HTML`, yielding the final `Main HTML`. For downstream usability, `Main HTML` is converted into Markdown format using the html2text[5] library.

## 3.2 PRE-PROCESSING AND POST-PROCESSING

`Raw HTML` is primarily designed for visual rendering, not for semantic interpretation by language models. Naively including all tags and attributes results in excessively long input sequences. Our pre-processing module is therefore guided by a multi-faceted strategy for simplification and chunking. The process begins with the **(1) preemptive removal of non-content tags**, such as `<style>`, `<script>`, `<header>`, and `<aside>`. Concurrently, we perform **(2) attribute simplification**, pruning all attributes except for class and id, which often carry the most valuable semantic cues for distinguishing content blocks. Following this, the document undergoes **(3) block-level chunking**, where it is segmented at elements that typically induce a line break in rendering. This strategy treats cohesive units like tables (`<table>`) and lists (`<ul>`) as indivisible blocks to preserve their integrity. To handle the common misuse of tables for page layout, we apply heuristic rules to permit splitting within them when necessary. Finally, to manage excessively long individual blocks, such as a table with many cells, a list with numerous items, or an overly long paragraph, we employ **(4) partial content truncation**. For instance, we may retain only a subset of table cells or the initial 200 characters of a long paragraph, as we empirically find this partial data is sufficient for accurate classification while significantly reducing input length.

This pre-processing pipeline transforms `Raw HTML` into a sequence of simplified blocks ready for Dripper-0.6B. To ensure the final output is a valid DOM subtree, the Mapping HTML is generated in parallel by applying only the block-level chunking to the original, unmodified HTML. The post-processing module then uses the Dripper-0.6B's output to select the corresponding content-bearing blocks from this `Mapping HTML` to construct the final result.

## 3.3 TASK FORMULATION

The system architecture detailed above effectively transforms the content extraction task into a well-defined **sequence labeling problem**. Formally, our pre-processing module converts an HTML doc-

---

[5]https://pypi.org/project/html2text/

ument into a sequence of $n$ simplified blocks, $X = [x_1, x_2, \ldots, x_n]$. Each block $x_i$ has a corresponding ground-truth label $y_i \in \{0, 1\}$, where 1 indicates main content and 0 indicates boilerplate. The core task is to train a model $f_\theta$ that takes the sequence $X$ as input and produces a predicted label sequence, $Y_{pred} = f_\theta(X)$, where $Y_{pred} = [y_1', y_2', \ldots, y_n']$. This predicted sequence is then used by the post-processing module to select the corresponding blocks from the `Mapping HTML` and construct the final `Main HTML`.

This sequence labeling formulation is highly efficient and reliable. By simplifying and chunking the input, the token load on the model is substantially reduced. Furthermore, framing the task as a classification of discrete blocks constrains the output to a simple sequence of binary labels. This design minimizes the required output length and, by avoiding free-form text generation, inherently eliminates the risk of hallucination, guaranteeing that the extracted content is a faithful subset of the original document.

### 3.4 Constrained Decoding via a Custom Logits Processor

To eliminate hallucination and guarantee a valid output format, we implement a custom logits processor that functions as a deterministic finite state machine (FSM) during decoding. The FSM precisely controls the generation of the JSON-like output structure (e.g., {"1": "main", ...}) by deterministically managing all syntactic tokens, such as braces, quotes, and numeric keys. At each decoding step, it masks the SLM's logits, permitting the model to make a probabilistic choice only at the single critical juncture of classifying a block. At this point, the vocabulary is restricted to just 'main' and 'other', effectively converting the task into a series of high-confidence binary classifications. This method guarantees syntactically perfect output, fundamentally removing the risk of format errors or extraneous content, and enables even a small 0.6B model to perform this structured prediction task with perfect fidelity.

## 4 Dataset and Benchmark

In this section, we detail the construction of our large-scale training dataset (Section 4.1) and our new evaluation benchmark, MainWebBench (Section 4.2), along with its evaluation metrics.

### 4.1 Training Data Construction

To train our model effectively, we construct a large-scale, multi-faceted training dataset engineered to capture the diversity of the modern web. The dataset is curated through a three-stage sampling and filtering pipeline, ensuring variety in page layout, language, and document format.

**Stage 1: Layout-Diverse Sampling.** The initial stage focuses on capturing structural diversity. We begin by grouping pages by domain across 107 dumps of the Common Crawl dataset. For each domain, we featurize the DOM tree structure of its pages (capped at 10,000 randomly sampled pages for larger domains) and computed their pairwise cosine similarity. We then apply the DBSCAN algorithm to these feature vectors to identify distinct layout clusters. From this process, we sample one representative webpage from each of approximately 40 million unique clusters, yielding a candidate pool of 40 million structurally diverse pages.

**Stage 2: Multilingual and Format-Aware Filtering.** From this candidate pool, the second stage filtered for linguistic and format diversity. We first extract the main content of each page using `Trafilatura` and then employ `Fasttext` lid-176[6] model for language identification. This step produced a balanced 10-million-page subset (4.75M English, 4.75M Chinese, 0.5M other languages). To further enhance diversity, we categorize these pages using the format classifier proposed by (Wettig et al., 2025a). A final balanced sampling across these identified formats results in a set of approximately 1 million pages (485k English, 487k Chinese, 50k other) for the final annotation stage.

**Stage 3: Final Annotation.** In the final stage, we process these 1 million pages through our simplification algorithm (detailed in Section 3.2). The resulting `Simplified HTML` is then provided to

---

[6]https://fasttext.cc/docs/en/language-identification.html

the Deepseek-chat API with a carefully crafted prompt (see Appendix Figure 6) to generate block-level labels. This automated pipeline yields approximately 1 million pages with high-quality, block-level annotations. After a final filtering step to remove samples containing no main content (i.e., all blocks were labeled as 'other'), we obtain our final training dataset of 870,945 samples.

## 4.2 MAINWEBBENCH: A NEW BENCHMARK FOR CONTENT EXTRACTION

To facilitate a more rigorous and fine-grained evaluation of web content extraction, we construct **MainWebBench**, a new benchmark comprising 7,887 meticulously annotated samples. Each sample contains four keys: 'html'( the raw html document); 'main_html'( the ground-truth as a valid html subtree identified by human annotators); 'convert_main_content'( a Markdown representation, generated from the ground-truth); and 'meta'( a rich set of annotations). MainWebBench is designed to serve as a gold-standard resource for evaluating extraction accuracy and enabling multi-dimensional performance analysis. An example data entry is shown in Appendix Figure 4.

### 4.2.1 BENCHMARK CONSTRUCTION

MainWebBench is constructed using a hybrid sampling strategy to ensure broad representation: 90% of pages are randomly sampled from Common Crawl to cover the long-tail of the web, while 10% are drawn from a list of top-ranking websites (Chinaz Alexa[7]) to include popular, well-designed pages. To address the ambiguity in defining "main content," we establish annotation rules based on two principles: **Contextual Integrity**, which includes content integral to the primary article (e.g., abstracts, references) and excludes peripheral elements (e.g., related-articles sidebars); and **Human-Generated Content**, which focuses on substantive material like article bodies and comments while filtering out auto-generated metadata (e.g., timestamps). Each page is meticulously annotated through a rigorous multi-stage process by using a custom-built tool( see Appendix Figure 3). Furthermore, we enrich the benchmark with rich metadata annotations—including language, style, a quantitative difficulty level, and rich content tags—enabling fine-grained analysis. More details of benchmark construction can be found in Appendix A.5

### 4.2.2 EVALUATION METRICS

To accommodate the two primary output formats of extraction tools—(1) raw Markdown text and (2) `Main HTML` document—we establish a standardized evaluation protocol. For the latter case, all `Main HTML` outputs are first converted to a canonical Markdown representation using the `html2text` library to ensure a fair and consistent comparison. The primary evaluation metric is the ROUGE-N F1 score, computed between the predicted Markdown and the ground-truth. We use the jieba tokenizer for all computations and set N=5. We specifically choose ROUGE-N instead of ROUGE-L, as the latter's Longest Common Subsequence (LCS) algorithm has prohibitive computational complexity on the long documents in our benchmark, making ROUGE-N a more scalable and practical choice for evaluation.

## 5 EXPERIMENTS

### 5.1 EXPERIMENTAL SETUP

**Supervised fine-tuning.** We employ the Qwen3-0.6B( (Team, 2025)) model as our base model, which is the smallest model in the Qwen3 series, featuring a 32K context window and support for over 100 languages. Supervised fine-tuning is performed using the Llama-Factory( (Zheng et al., 2024)) framework, training on the full set of 870K samples for a fixed total of 4 epochs. We use the last checkpoint as **Dripper-0.6B**.

**Baseline Methods.** To comprehensively evaluate Dripper, we compare it against a diverse set of establish and state-of-the-art content extraction systems. Our comparison spans a wide spectrum of approaches, including classic heuristic and rule-based systems, supervised learning methods, production-grade hybrid tools, and recent large language model-based extractors. A detailed list and description of each baseline method is provided in Appendix, Table 4.

---

[7]https://malexa.chinaz.com/

**Evaluation Modes.** To ensure a fair comparison across tools with diverse output capabilities, we established a clear evaluation protocol. We test every applicable output format for each tool and use a consistent suffix to denote the mode: `-HTML+MD` for tools that output an intermediate HTML which we convert to Markdown; `-MD` for tools that natively output Markdown; and `-TEXT` for tools that natively output plain text. Because Dripper cannot process inputs that exceed its context-length limit, we assign a score of 0 to such inputs. Following the practice of `Trafilatura`, which uses a fallback algorithm for parsing failures, we also test a version of our method, Dripper_fallback, which invokes `Trafilatura` for oversized inputs.

## 5.2 RESULT OF OVERHEAD REDUCTION

The computational cost of a decoder-only language model is primarily determined by the input and output sequence lengths, with its complexity approximated by Equation (1).

$$\text{Cost} \approx \left( \text{L d} \left( \text{N}^2 + \text{M N} + \text{M}^2 \right) + \text{L d}^2 \left( \text{N} + \text{M} \right) \right) \text{flops} \tag{1}$$

where $L$ is the number of attention layers, $d$ is the hidden-state dimension, $N$ is the number of input tokens, and $M$ is the number of output tokens. For Qwen3-0.6B we set $L = 28$ and $d = 1024$.

To quantify the efficiency gains of our approach, we compare its cost against a naive generative baseline. The baseline cost is estimated by using `Raw HTML` as input to generate the full Markdown content. For our method, we use `Simplified HTML` as input and the structured JSON classification as output. We measure the token lengths for both scenarios on the MainWebBench, and the results are detailed in Table 1.

Table 1: Computational overhead comparison on MainWebBench. The Pre-process column distinguishes the methods: Without denotes the naive baseline (generating full Markdown from Raw HTML), while With denotes the Dripper framework (predicting JSON labels from Simplified HTML). We report mean and median values for Input/Output token lengths and estimated inference cost (FLOPs), with the Ratio row demonstrating the efficiency gains of Dripper.

| Pre-process | Input length (tokens) | | Output length (tokens) | | Cost estimate (flops) | |
|---|---|---|---|---|---|---|
| | mean | median | mean | median | mean | median |
| Without | 44705.9 | 31987.0 | 2303.7 | 675.0 | $1.102 \times 10^{14}$ | $3.206 \times 10^{13}$ |
| With | 5734.5 | 3109.0 | 383.4 | 187.0 | $5.702 \times 10^{12}$ | $5.254 \times 10^{11}$ |
| Ratio | 12.83% | 9.72% | 16.64% | 27.70% | 5.18% | 1.64% |

The results reveal a substantial reduction in computational overhead. Our pre-processing pipeline dramatically shortens the input, reducing the mean token count to just 12.83% of `Raw HTML`, which is crucial for fitting within the model's context window. Simultaneously, reframing the task to output a compact JSON classification reduces the mean output length to 16.64% of the full content. These two synergistic effects culminate in a remarkable reduction in computational load, lowering the mean inference cost to just 5.18% of the naive approach. This makes SLM-based content extraction not only feasible but also highly efficient and controllable.

## 5.3 RESULTS ON MAINWEBBENCH

We present the main performance comparison on our MainWebBench benchmark in Table 2. The results are broken down by various tracks, including difficulty levels and the presence of rich content.

The results clearly demonstrate that Dripper achieves state-of-the-art performance, significantly outperforming all baseline methods across every track. The standalone Dripper model achieves an overall score of 0.8182, surpassing the best baseline, magic-html (0.7091), by a large margin. Notably, Dripper shows exceptional strength on challenging content types where traditional methods falter, such as pages with tables, equations, and especially conversational layouts (0.8028 vs. 0.5766 for the best baseline). This highlights the robustness of our semantic, block-based classification approach.

Table 2: Performance comparison on MainWebBench (ROUGE-N F1). Methods are categorized by Mode: HTML+MD denotes tools outputting intermediate HTML converted to Markdown, while MD and TEXT denote native Markdown and Plain Text outputs, respectively. Results are stratified by Overall performance, Difficulty Level (simple, mid, hard), and specific Rich Content Types (subsets containing tables, code, equations, or conversational text).

| name | mode | all | simple | mid | hard | table | code | equation | conversational |
|---|---|---|---|---|---|---|---|---|---|
| magic-html (opendatalab, 2024) | Html+MD | 0.7091 | 0.7811 | 0.7095 | 0.6367 | 0.6681 | 0.8471 | 0.8470 | 0.4678 |
| Readability (Mozilla, 2015) | Html+MD | 0.6491 | 0.7370 | 0.6525 | 0.5570 | 0.5896 | 0.7774 | 0.7800 | 0.4608 |
| Trafilatura (Barbaresi, 2021) | Html+MD | 0.6358 | 0.7277 | 0.6391 | 0.5396 | 0.5505 | 0.6006 | 0.7327 | 0.5750 |
| Resiliparse (Bevendorff et al., 2018) | TEXT | 0.6233 | 0.7099 | 0.6283 | 0.5304 | 0.5473 | 0.6474 | 0.7829 | 0.5346 |
| Trafilatura | MD | 0.6237 | 0.7115 | 0.6279 | 0.5305 | 0.5400 | 0.5741 | 0.7168 | 0.5766 |
| Trafilatura | TEXT | 0.6049 | 0.6900 | 0.6088 | 0.5149 | 0.5271 | 0.5566 | 0.6955 | 0.5681 |
| html2text (Swartz et al., 2025) | MD | 0.5977 | 0.7499 | 0.5812 | 0.4678 | 0.5937 | 0.7729 | 0.7129 | 0.5494 |
| BoilerPy3 (Riebold et al., 2023) | TEXT | 0.5413 | 0.6347 | 0.5448 | 0.4434 | 0.4380 | 0.4833 | 0.6590 | 0.4695 |
| GNE (Kingname et al., 2024) | Html+MD | 0.5148 | 0.6477 | 0.4942 | 0.4098 | 0.4129 | 0.5495 | 0.6160 | 0.3296 |
| news-please (Hamborg et al., 2017) | TEXT | 0.5012 | 0.5399 | 0.5250 | 0.4307 | 0.4193 | 0.5118 | 0.6701 | 0.4073 |
| jusText (Pomikálek, 2011) | TEXT | 0.4770 | 0.5132 | 0.5070 | 0.4010 | 0.3962 | 0.3779 | 0.6652 | 0.5222 |
| BoilerPy3 | Html+MD | 0.4766 | 0.6443 | 0.4706 | 0.3174 | 0.3783 | 0.5532 | 0.6157 | 0.4103 |
| Goose3 (Lababidi et al., 2025) | TEXT | 0.4354 | 0.4514 | 0.4645 | 0.3808 | 0.3589 | 0.2900 | 0.6376 | 0.3064 |
| ReaderLM-v2 (Wang et al., 2025) | MD | 0.2264 | 0.3374 | 0.2078 | 0.1403 | 0.1801 | 0.2431 | 0.2927 | 0.1537 |
| Dripper | Html+MD | **0.8182** | **0.8837** | **0.8178** | **0.7536** | **0.7693** | **0.8368** | **0.8889** | **0.7671** |
| Dripper_fallback | Html+MD | **0.8399** | **0.9010** | **0.8392** | **0.7799** | **0.7964** | **0.8673** | **0.9067** | **0.8028** |

Additionally, due to limitations in preprocessing capacity and model generalization, Dripper occasionally fails to extract meaningful content from certain pages. We note that since Dripper follows a fundamentally different technical approach compared to rule-based systems like `Trafilatura`, its failures tend to be orthogonal to those of such systems. This allows for a straightforward fallback strategy: when Dripper returns no valid output, we use `Trafilatura` as a backup. With this mechanism, the combined system (Dripper_fallback) achieves an overall F1 score of 0.8399. This result indicates that our semantic approach not only establishes a new state-of-the-art on its own but can also be effectively combined with existing methods to improve robustness and coverage.

## 5.4 ABLATION STUDY

To analyze the data efficiency of our approach, we fine-tune the Qwen3-0.6B model on training subsets of increasing size: 2k, 5k, 10k, 100k, and 870k. We evaluate each resulting checkpoint on MainWebBench, from which we excluded samples whose simplified HTML exceeded our 32k token context window, as the standard Dripper model is designed to score 0 on such oversized inputs. This results in a performance gap of about 1.9% (0.818 for the full bench and 0.834 for the filtered bench).

To isolate the impact of our constrained decoding mechanism, we compare the performance of models trained with and without the custom logits processor. As shown in Figure 2, the logits processor provides a consistent performance improvement across nearly all data scales. The most significant gain (+2.3%) is observed at the 2k data scale, indicating that the FSM provides a strong structural prior that helps the model learn the task more efficiently in low-data regimes. As the training set grows, the model begins to learn the output format implicitly, and the performance gap narrows. Nevertheless, the logits processor provides an absolute guarantee of a syntactically perfect, hallucination-free output. This ensures the output is always stable and machine-readable, preventing format errors that would otherwise disrupt downstream tasks

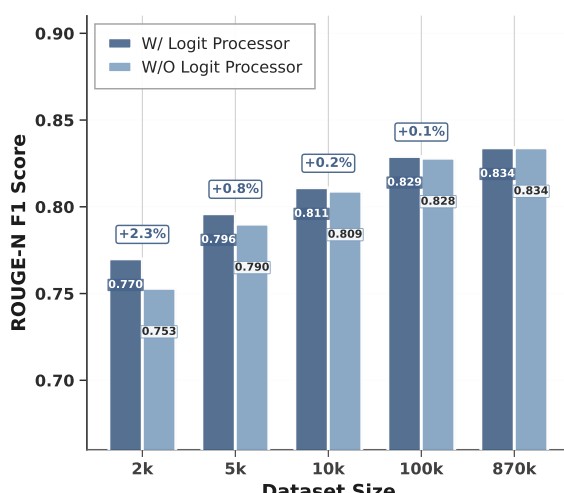

Figure 2: Impact of the logits processor on performance across various training data scales.

and making the processor a critical component for production-level reliability.

## 5.5 PERFORMANCE ON WCEB

To assess the generalization capabilities of Dripper, we evaluate it on the Web Content Extraction Benchmark (WCEB, (Bevendorff et al., 2023)) , a comprehensive and unified benchmark. WCEB addresses inconsistencies prevalent in many legacy datasets—such as plain-text-only ground truths, file encoding errors, and corrupted content from script injections—by providing a filtered and standardized collection. Since the ground truths in this consolidated benchmark are in plain text, we adapt our evaluation protocol by using the `html-text`[8] library for the final conversion, a configuration we denote as `Html+TEXT`. To enable a more granular analysis, we also apply our difficulty stratification scheme to this dataset. A detailed description of the benchmark can be found in Appendix, Table 5.

The results on this suite of nine established benchmarks, presented in Table 3, confirm Dripper's strong generalization capabilities. Our method again establishes a new state-of-the-art, with the standalone Dripper model (0.8002) outperforming the strongest prior method, `Trafilatura` (0.7833). Furthermore, echoing the findings on MainWebBench, the Dripper_fallback strategy again demonstrates the complementary nature of our SLM-based approach and traditional heuristics, boosting the score further to 0.8154. This strong performance across a diverse collection of legacy datasets highlights Dripper's robustness, setting a new state-of-the-art for general web content extraction.

Table 3: Generalization performance on the WCEB dataset (ROUGE-N F1). Given the benchmark's plain-text ground truth, Mode is defined as: Html+TEXT (converting extracted HTML to text) or TEXT (native text output). Results are stratified by Overall performance and Difficulty Level (simple, mid, hard) to demonstrate model robustness across varying page complexities.

| name | mode | all | simple | mid | hard |
|---|---|---|---|---|---|
| Trafilatura | TEXT | 0.7833 | 0.8122 | 0.7785 | 0.7609 |
| Trafilatura | Html+TEXT | 0.7791 | 0.7896 | 0.7758 | 0.7731 |
| Readability | Html+TEXT | 0.7642 | 0.7744 | 0.7595 | 0.7601 |
| magic-html | Html+TEXT | 0.7506 | 0.7780 | 0.7573 | 0.7144 |
| Goose3 | TEXT | 0.7272 | 0.7432 | 0.7312 | 0.7059 |
| Resiliparse | TEXT | 0.7225 | 0.7697 | 0.7052 | 0.6985 |
| news-please | TEXT | 0.7048 | 0.7051 | 0.7103 | 0.6970 |
| justText | TEXT | 0.6936 | 0.7445 | 0.6966 | 0.6389 |
| BoilerPy3 | TEXT | 0.6221 | 0.6481 | 0.6468 | 0.5631 |
| html2text | TEXT | 0.6142 | 0.7273 | 0.6165 | 0.4982 |
| BoilerPy3 | Html+TEXT | 0.6015 | 0.6532 | 0.6035 | 0.5474 |
| GNE | Html+TEXT | 0.5166 | 0.5138 | 0.5069 | 0.5323 |
| ReaderLM-v2 | TEXT | 0.3077 | 0.3718 | 0.2928 | 0.2636 |
| Dripper | Html+TEXT | **0.8002** | **0.8293** | **0.8005** | **0.7707** |
| Dripper_fallback | Html+TEXT | **0.8154** | **0.8363** | **0.8143** | **0.7959** |

## 6 CONCLUSION

In this work, we introduce Dripper, a highly efficient and accurate framework for web content extraction. We demonstrate that our custom-trained 0.6B parameter Small Language Model, Dripper-0.6B, achieves state-of-the-art performance by reframing the extraction problem. Our approach's success is rooted in three key technical contributions. First, our HTML Simplification Algorithm intelligently strips redundant tags and attributes, drastically reducing the input token count while preserving essential structural cues. This simplified document is then processed through our novel Sequential Block Classification paradigm, which transforms the open-ended extraction task into a series of simple, localized binary classifications. Finally, to guarantee absolute fidelity, our Deterministic Logits Processor constrains the SLM's output during the decoding phase, which completely eliminates the risk of hallucination and ensures a syntactically perfect structured output. To rigorously validate our method, we also construct and release MainWebBench, a new large-scale benchmark of 7,887 samples, on which Dripper-0.6B proves its superiority over all baselines. Furthermore, by integrating a heuristic-based fallback for inputs that exceed its context window, our Dripper_fallback variant pushes performance even higher, demonstrating the robustness and complementary nature of our method.

---

[8]https://pypi.org/project/html-text/

## 7 LIMITATION AND FUTURE WORK

Despite careful web preprocessing development, 1.3% of Common Crawl pages still exceed Qwen3's content-window limit post-simplification and remain unprocessable. Additionally, extreme DOM structures in some pages break chunking/simplification algorithms, hindering effective main text extraction. Future fixes include enhancing preprocessing and extending the base model's context window via continued pre-training (to relax preprocessing's token budget). Moreover, while we use Qwen3's smallest 0.6B model to cut overhead, scaling to 100B-scale pages poses cost issues. A promising solution is tailoring data recipes for web parsing to pre-train small (0.01B–0.1B) dedicated base models from scratch, lowering inference costs.

## 8 REPRODUCIBILITY STATEMENT

We are committed to ensuring the full reproducibility of our research. The architecture of our proposed framework, **Dripper**, and its core components are detailed in the Methodology Section 3. The construction of our large-scale training dataset is described in Section 4.1, while the creation and structure of our new benchmark are detailed in the **MainWebBench** Section 4.2. Our complete experimental setup, including all baselines, evaluation protocols, and metrics, is presented in the Experiments Section 5. To facilitate direct verification and future work, we have made our resources publicly available: the full source code[9], the trained **Dripper** model weights[10], and the complete **MainWebBench** benchmark[11].

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

# A    APPENDIX

## A.1    BASELINE METHODS FOR WEB CONTENT EXTRACTION

Table 4: An overview of the baseline methods for web content extraction.

| Method | Description |
|---|---|
| ***Heuristic and Rule-Based Methods*** | |
| Readability | Reader view algorithm for removing distracting elements |
| jusText | Two-pass processing with block size, link density, and stopword heuristics |
| Goose3 | Article extractor with hand-crafted rules |
| html2text | Simple HTML to markdown converter |
| GNE | Text and symbol density-based extraction using mathematical formulas |
| Resiliparse | Fast and robust heuristic extractor with HTML parsing |
| ***Supervised Learning Methods*** | |
| BoilerPy3 | Python port of Boilerpipe, decision tree-based text block classification |
| ***Hybrid Systems and Production Tools*** | |
| Trafilatura | Sophisticated rule cascade with jusText and Readability as fallbacks |
| news-please | Meta-extractor combining multiple extractors for news articles |
| magic-html | HTML structure simplification for extraction pipelines |
| ***Pre-trained Language Models*** | |
| ReaderLM-v2 | SLM-based content extraction with semantic understanding |

## A.2    STANDARD BENCHMARKS

Table 5: Details of the datasets that comprise the Web Content Extraction Benchmark (WCEB).

| Dataset | Pages | Source & Characteristics |
|---|---|---|
| CleanEval | 738 | De-facto standard dataset from 2007 shared task combining development and evaluation sets of English web pages with basic structural markup ground truth |
| CleanPortalEval | 71 | Extension of CleanEval featuring multi-page samples from 4 major news domains |
| CETD | 700 | Created for density-based extractor evaluation across 6 domains |
| Dragnet | 1,379 | Combined sources from popular RSS feeds, 23 major news sites, 178 Technorati blogs, plus CETR and CleanEval conversions |
| L3S-GN1 | 621 | Created by BoilerPipe authors with unique HTML annotation using span-wrapped CSS classes for 5-level content relevance |
| Google-Trends-2017 | 180 | Dataset created for BoilerNet neural network training featuring binary CSS class annotations on DOM leaf nodes to distinguish content from boilerplate |
| Readability | 115 | Mozilla reader mode test suite with original and simplified HTML for evaluation |
| Scrapinghub | 181 | Created by Zyte for benchmarking proprietary extraction services |

## A.3 SCREENSHOT OF THE WEB PAGE ANNOTATION TOOL

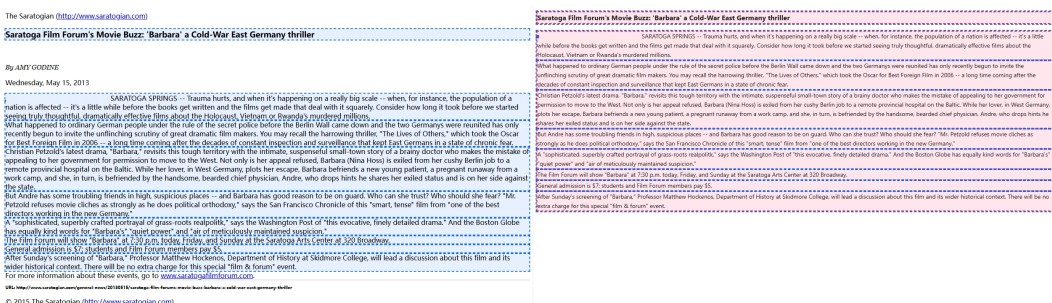

Figure 3: Screenshot of the web page annotation tool. The main content selection is highlighted in blue on the left, with a real-time preview on the right.

## A.4 EXAMPLE DATA FROM MAINWEBBENCH

```
1  {
2    "track_id": "XXXX",
3    "html": "<html><body><h1 cc-select=True>Hello
     ↪   world!</h1><aside>advertisement</aside></body></html>",
4    "main_html": "<html><body><h1>Hello world!</h1></body></html>",
5    "convert_main_content": "# Hello world!",
6    "meta": {
7      "language": "en",
8      "style": "Normal",
9      "level": "easy",
10     "table": "without",
11     "code": "without",
12     "equation": "without"
13   }
14 }
```

Figure 4: An example data from MainWebBench. It includes the raw source, the ground-truth main HTML, its Markdown conversion, and a rich set of metadata for fine-grained analysis.

## A.5 BENCHMARK CONSTRUCTION

**Data Sampling.** MainWebBench is constructed using a hybrid sampling strategy to ensure both broad representation and relevance. 90% of the samples are randomly drawn from the Common Crawl dataset to cover the long-tail web, while the remaining 10% are sampled from a list of top-ranking websites (Chinaz Alexa[12]) to include popular, professionally designed pages. The final benchmark is highly diverse, containing pages from 5,434 unique top-level and 5,904 unique second-level domains.

**Annotation Rules.** To address the ambiguity in defining "main content" for unconventional layouts, we establish two core annotation principles. First, **Contextual Integrity** dictates that content integral to the main article—such as a table of contents, abstract, or reference list—is included. Conversely, contextually independent elements like "related articles" sidebars or copyright footers are excluded. Second, the main content is defined as **Human-Generated Content**, including article bodies, user comments, and Q&A posts, while associated auto-generated metadata like usernames and timestamps are excluded.

**Annotation Process.** The annotation for each page followed a rigorous three-stage process using a custom-built tool(see Appendix, Figure 3) that allowed for tag-level granularity. The process

---

[12]https://malexa.chinaz.com/

involved: (1) an initial pass by one annotator, (2) a review and correction pass by a second annotator, and (3) a final quality assurance check by a senior inspector, who made the final adjudication to resolve any discrepancies. Pages uninterpretable due to rendering issues were discarded.

**Metadata Annotation.** To enable detailed, fine-grained analysis, we annotate each page with a rich set of metadata. This includes **Language**, identified by GPT-5(OpenAI, 2025) and labeled as `en` (English) or `non_en` (other), and **Style**, classified by GPT-5 as `Conversational` for pages with user-generated content or `Normal` otherwise. We also develop a quantitative **Difficulty Level**, determined by an `overall_complexity_score` calculated for each page. To compute this score, we first measure four distinct metrics: *DOM structural complexity* (based on tree depth and width), *text distribution sparsity* (transitions between text/non-text nodes), *content-type diversity* (a count of rich content types), and *link density* (the ratio of hyperlinked text). These four values are individually normalized, and their weighted sum produces the final score. Based on the distribution of this `overall_complexity_score` across the benchmark, we then categorize pages into `simple`, `medium`, and `hard` using the 30th and 70th percentiles as dynamic thresholds. Finally, we add **Rich Content Tags** to identify the presence of tables (`<table>`), code blocks (``), and mathematical formulas (`<math>` or LaTeX patterns) using BeautifulSoup.

## A.6 DETAILED BENCHMARK STATISTICS

In this section, we provide granular statistics regarding the composition of MainWebBench to demonstrate its diversity and coverage. MainWebBench consists of 7,887 samples. As detailed in Section 4.2.1, the composition follows a hybrid sampling strategy: 90% are randomly sampled from Common Crawl to capture the "long-tail" of the web, while 10% are sampled from top-ranking websites to ensure the inclusion of popular, high-quality pages.

**Domain Diversity.** The dataset covers 5,945 unique domains, confirming that the data is not dominated by a few sources but possesses a high degree of diversity. Table 6 lists the top 10 domains sorted by sample count. Furthermore, the benchmark spans 150 distinct Top-Level Domains (TLDs), indicating a broad spectrum of global regions and website categories. The distribution of the top 10 TLDs is presented in Table 7.

**Page Category Distribution.** We utilized GPT-5 to classify the semantic type of every page in the benchmark. As visually demonstrated in Figure 5, the dataset covers a diverse range of page layouts, ranging from standard news articles to forums and product pages.

**Language Diversity.** The dataset includes web pages in 46 different languages. We present the partial language statistics (top 10) in Table 8. The complete statistical files have been uploaded to Hugging Face[13].

Table 6: Partial Domain Statistics (Top 10 Sorted by Sample Count). This table highlights the variety in page styles, difficulty levels, and rich content elements even within the most frequent domains.

| Domain | Count | Percent | Lang | Style | Level | Table | Code | Eq. |
|---|---|---|---|---|---|---|---|---|
| aniruddhadeb.com | 39 | 0.49% | en | Article | simple | 1 | 9 | 36 |
| politics.stackexchange.com | 30 | 0.38% | en | Forum | mid | 0 | 0 | 0 |
| www.ask.com | 29 | 0.37% | en | Article | simple | 1 | 0 | 3 |
| en.wikipedia.org | 27 | 0.34% | en | Article | hard | 20 | 1 | 0 |
| www.china.org.cn | 23 | 0.29% | en | Article | simple | 21 | 0 | 0 |
| money.cnn.com | 22 | 0.28% | en | Article | hard | 18 | 0 | 7 |
| data.epo.org | 21 | 0.27% | en | Article | simple | 21 | 0 | 0 |
| m.weibo.cn | 19 | 0.24% | zh | Forum | simple | 0 | 0 | 0 |
| spanish.china.org.cn | 15 | 0.19% | es | Article | simple | 14 | 0 | 0 |
| china.org.cn | 14 | 0.18% | en | Article | mid | 13 | 0 | 0 |

## A.7 PROMPT FOR DATA SYNTHESIS

---

[13]`https://huggingface.co/anonymous-s2wrvq/Dripper`

Table 7: Partial Top-Level Domain (TLD) Distribution (Top 10).

| TLD | Count | Percent |
|---|---|---|
| com | 4550 | 57.69% |
| org | 816 | 10.35% |
| cn | 459 | 5.82% |
| net | 318 | 4.03% |
| uk | 235 | 2.98% |
| edu | 180 | 2.28% |
| de | 101 | 1.28% |
| au | 94 | 1.19% |
| ru | 69 | 0.87% |
| gov | 59 | 0.75% |

Table 8: Partial Language Distribution (Top 10) in the benchmark.

| Language | Count | Percent |
|---|---|---|
| English | 6711 | 85.09% |
| Chinese | 716 | 9.08% |
| Spanish | 61 | 0.77% |
| German | 51 | 0.65% |
| Japanese | 48 | 0.61% |
| Russian | 45 | 0.57% |
| French | 36 | 0.46% |
| Italian | 22 | 0.28% |
| Korean | 20 | 0.25% |
| Portuguese | 17 | 0.22% |

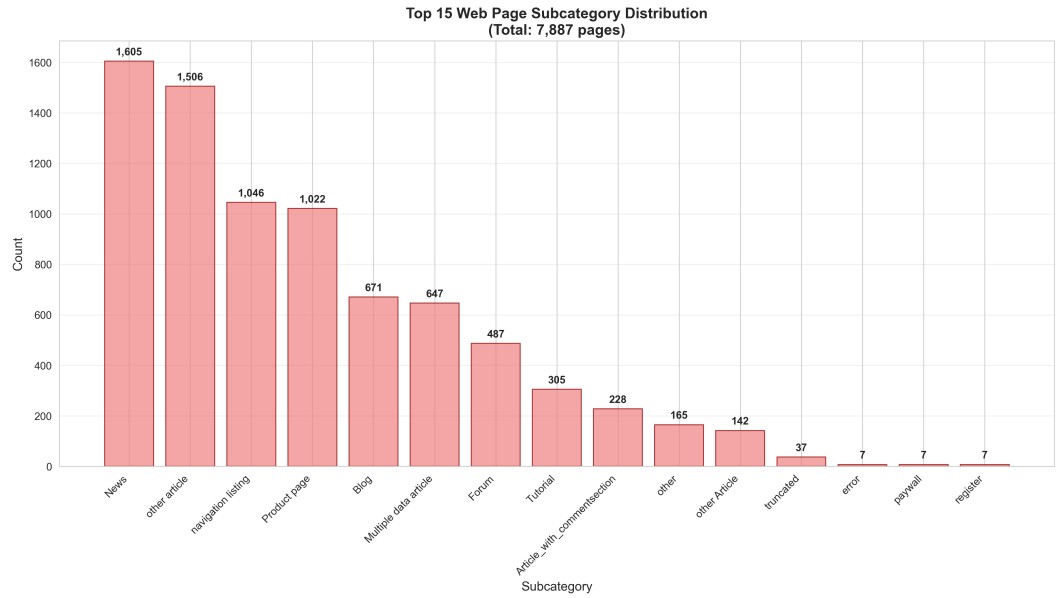

Figure 5: Top 15 Web Page Subcategory Distribution. The types were classified semantically using GPT-5. The distribution shows a wide coverage from standard articles to forums and product pages.

## A.8 ANALYSIS OF CLASSIFICATION METRICS

In this section, we provide a detailed analysis of the model's internal classification performance relative to the final extraction quality. In fact, we closely monitored these metrics (Precision, Recall, and F1) throughout our model development process to assess internal classification performance. We provide these results in Table 9.

Table 9: Block-level classification metrics and ROUGE-N F1 across different training data sizes.

| Data Size | Block-level Precision | Block-level Recall | Block-level F1 | ROUGE-N F1 |
|---|---|---|---|---|
| 2k | 0.877 | 0.781 | 0.756 | 0.770 |
| 5k | 0.875 | 0.810 | 0.781 | 0.796 |
| 10k | 0.888 | 0.823 | 0.800 | 0.811 |
| 100k | 0.900 | 0.838 | 0.821 | 0.829 |
| 870k | 0.898 | 0.843 | 0.826 | 0.834 |

While the data confirms our model's strong classification capability, we deliberately chose ROUGE-N as our primary reporting metric for three key reasons.

First, our pre-processing creates blocks with significant content length variance. A block can range from a single boilerplate word to a 2,000-word main article. Standard classification metrics treat all blocks equally, meaning a model could achieve a high F1 score by correctly classifying hundreds of tiny boilerplate blocks while missing the single, massive main content block. This would yield a high classification score but a completely failed extraction.

Secondly, ROUGE-N better aligns with the end-user's objective, which is to obtain the complete main text. By measuring the overlap between the extracted text and the ground truth, ROUGE implicitly weights blocks by their information content, ensuring that the metric reflects the actual utility of the output.

Finally, prioritizing ROUGE-N ensures consistency with established benchmarks in the web extraction literature, where ROUGE-L or ROUGE-N are the standard metrics for comparison.

### A.9 PERFORMANCE COMPARISON OF LLM AND DRIPPER

We compared performance of Dripper with GPT-5 (OpenAI, 2025), DeepSeek-V3.2-Exp (DeepSeek-AI, 2025) and Claude-Sonnet-4-5-20250929 (Anthropic, 2025) on the same input. Notably, our 0.6B Dripper model (0.8182) achieves 98.4% of the performance level of the state-of-the-art Claude-Sonnet-4.5 (0.8319). Although frontier LLMs exhibit a slight advantage in handling complex formatting tasks such as equations and conversational content, our Dripper_fallback strategy effectively bridges this gap, achieving an overall F1 score of 0.8399 that surpasses even the best-performing frontier models. Crucially, Dripper delivers this SOTA-level performance using a lightweight, locally deployable model, thereby avoiding the prohibitive latency and costs associated with querying massive frontier models for web-scale extraction.

Table 10: Performance Comparison of LLM and Dripper

| Model | All | Simple | Mid | Hard | Table | Code | Equation | Conversational |
|---|---|---|---|---|---|---|---|---|
| GPT-5 | 0.8302 | 0.8815 | 0.8301 | 0.7792 | 0.7957 | 0.8707 | 0.9161 | 0.7992 |
| DeepSeek-V3 | 0.8252 | 0.8826 | 0.8244 | 0.7690 | 0.7804 | 0.8440 | 0.9113 | 0.8160 |
| Claude-Sonnet-4.5 | 0.8319 | 0.8890 | 0.8329 | 0.7737 | 0.7919 | 0.8619 | 0.9273 | 0.8062 |
| Dripper | 0.8182 | 0.8837 | 0.8178 | 0.7536 | 0.7693 | 0.8368 | 0.8889 | 0.7671 |
| Dripper_fallback | 0.8399 | 0.9010 | 0.8392 | 0.7799 | 0.7964 | 0.8673 | 0.9067 | 0.8028 |

### A.10 DETAILED PRE-PROCESSING ALGORITHM

In this section, we provide a comprehensive description of the HTML simplification algorithm, which serves as the cornerstone of the Dripper framework. The primary goal of this algorithm is to **drastically reduce the HTML token count while preserving the critical semantic and structural cues necessary for accurate content classification**. This is achieved through a multi-stage process applied to the raw HTML.

**1. DOM Cleaning and Pruning.** We first parse the HTML and proactively remove entire subtrees known to be boilerplate. This includes tags such as `<script>`, `<style>`, `<header>`, `<footer>`, and `<nav>`. Furthermore, we heuristically remove elements whose `class` or `id` attributes contain keywords like 'nav', 'footer', or 'header', or which have CSS styles indicating they are hidden (e.g., `display: none`).

**2. Attribute Simplification.** To reduce noise and token overhead, we strip nearly all attributes from all elements. The only exceptions are the `class` and `id` attributes, which are often the most informative semantic markers in modern web design, and for `` tags, we also preserve the `src` (excluding large base64 data) and `alt` attributes.

**3. Semantic Block Segmentation.** The core of our method involves converting the cleaned DOM tree into a linear sequence of semantic blocks. We perform a recursive traversal of the DOM, segmenting it at natural block-level boundaries. Our algorithm intelligently handles mixed content:

1. It identifies and preserves atomic block-level elements (e.g., a standalone paragraph or `<div>`).

2. It aggregates consecutive inline elements (e.g., ``, links with text) and unwrapped text nodes into coherent blocks, wrapping them in a custom tag if necessary to maintain structure.

3. It makes special provisions for complex structures like tables and lists, ensuring they are treated as single, indivisible units where appropriate.

**4. Content Truncation within Blocks.** To handle excessively long blocks (e.g., a massive list or a very long paragraph), we apply a conservative truncation strategy. We recursively traverse the block's content, limiting the total plain text to a predefined maximum length (e.g., 200 characters) while meticulously preserving the overall HTML tag structure. This ensures the model receives a representative sample of the content for classification without being overwhelmed by length.

**Parallel Generation Strategy.** A critical innovation in our pipeline is the parallel generation of `Simplified HTML` and `Mapping HTML`.Both representations undergo identical block segmentation, ensuring a one-to-one correspondence between blocks. However, `Simplified HTML` used for model input undergoes the full pruning and truncation process (steps 1-4). In contrast, `Mapping HTML`, used for final output reconstruction, undergoes only the initial cleaning (step 1) and segmentation (step 3), preserving the original, un-truncated

Finally, we inject a unique `_item_id` attribute to each block in both the Simplified and Mapping HTML. This allows the classification labels produced by Dripper-0.6B on the simplified sequence to be precisely mapped back to the rich, original content blocks for the final extraction.

### A.11    TRAINING CONFIGURATIONS

The specific hyperparameters and training configurations for supervised fine-tuning model are shown in table 11

Table 11: Supervised Fine-tuning (SFT) Configuration Details

| Category | Parameter | Value |
|---|---|---|
| Model & Framework | Base Model
Fine-tuning Method
Training Framework | Qwen3-0.6B
Full-parameter SFT
LLaMA-Factory(Zheng et al., 2024) |
| Training Dynamics | Epochs
Global Batch Size
Max Sequence Length | 4
128
32,000 |
| Optimizer & Scheduler | Learning Rate Scheduler
Peak Learning Rate
Warmup Ratio | Cosine
1.00E-04
0.1 |
| Hardware & Efficiency | Hardware
Precision | 32× NVIDIA A100 GPU
BF16 |

### A.12    DOMAIN-SPECIFIC EVALUATION RESULTS

To comprehensively assess the generalization capabilities of our framework, we utilized the Topic and Format classifiers proposed by Wettig et al. (2025b) to categorize all 7,887 samples in the MainWebBench. Based on this classification, we calculated the ROUGE-N F1 scores for all methods across 24 distinct topics (e.g., Science & Tech, Finance, Health) and 24 distinct formats (e.g., News Article, Tutorial, Forum).

As shown in Table 12, Dripper_fallback consistently secures the 1st place ranking **across various topic and format categories**. On average, it outperforms the strongest existing baseline (magic-html) by approximately **19%**, demonstrating the exceptional robustness of this strategy. The standalone Dripper model consistently ranks **second**.

These results consistently demonstrate our model's strong capabilities across diverse domains. The complete breakdown of performance scores for every individual Topic and Format category is presented in Tables 13 - 16.

Table 12: Average ROUGE-N F1 Scores and Rankings across Topics and Formats. Dripper and its fallback variant consistently outperform all baselines.

| (a) Topic Classification | | | (b) Format Classification | | |
|---|---|---|---|---|---|
| **Method** | **Avg Score** | **Rank** | **Method** | **Avg Score** | **Rank** |
| Dripper_fallback | 0.8583 | 1 | Dripper_fallback | 0.8320 | 1 |
| Dripper | 0.8330 | 2 | Dripper | 0.8102 | 2 |
| magic-html | 0.7209 | 3 | magic-html | 0.6974 | 3 |
| readability | 0.6829 | 4 | readability | 0.6342 | 4 |
| trafilatura-html-md | 0.6725 | 5 | trafilatura-html-md | 0.6259 | 5 |
| resiliparse | 0.6672 | 6 | resiliparse | 0.6122 | 7 |
| trafilatura-md | 0.6596 | 7 | trafilatura-md | 0.6144 | 6 |
| trafilatura-text | 0.6413 | 8 | trafilatura-text | 0.5956 | 8 |
| html2text-md | 0.6185 | 9 | html2text-md | 0.5827 | 9 |
| boilerpy3-text | 0.5811 | 10 | boilerpy3-text | 0.5377 | 10 |
| newsplease | 0.5595 | 11 | newsplease | 0.4944 | 12 |
| gne | 0.5499 | 12 | gne | 0.5049 | 11 |
| justtext | 0.5412 | 13 | justtext | 0.4766 | 13 |
| boilerpy3-html-md | 0.5229 | 14 | boilerpy3-html-md | 0.4676 | 14 |
| goose3 | 0.4775 | 15 | goose3 | 0.4321 | 15 |
| readerlm | 0.2492 | 16 | readerlm | 0.2205 | 16 |

Table 13: Detailed ROUGE-N F1 Scores across Topics (Part 1/2).

| Method | Home & Hobbies | Politics | Education | Software | Crime & Law | Science & Tech | Food & Drink | Social Life | Sports & Fitness | History | Finance | Literature |
|---|---|---|---|---|---|---|---|---|---|---|---|---|
| Dripper_fallback | 0.7971 | 0.9119 | 0.8543 | 0.7890 | 0.8956 | 0.8519 | 0.8654 | 0.8547 | 0.8373 | 0.8334 | 0.8505 | 0.8107 |
| Dripper | 0.7993 | 0.8886 | 0.8301 | 0.7730 | 0.8558 | 0.8281 | 0.8452 | 0.8337 | 0.8210 | 0.7884 | 0.8340 | 0.7667 |
| magic-html | 0.6176 | 0.8139 | 0.7093 | 0.6505 | 0.8140 | 0.7433 | 0.6673 | 0.7102 | 0.7035 | 0.6983 | 0.7578 | 0.6751 |
| readability | 0.5147 | 0.7899 | 0.6439 | 0.6167 | 0.7555 | 0.6851 | 0.5963 | 0.6302 | 0.6464 | 0.6438 | 0.6922 | 0.6284 |
| trafilatura-html | 0.5430 | 0.7709 | 0.6344 | 0.6140 | 0.7475 | 0.6621 | 0.6360 | 0.6758 | 0.6129 | 0.6424 | 0.6657 | 0.6492 |
| resiliparse | 0.5570 | 0.7402 | 0.6238 | 0.5858 | 0.7341 | 0.6599 | 0.5997 | 0.6144 | 0.6058 | 0.6171 | 0.6420 | 0.5947 |
| trafilatura-md | 0.5283 | 0.7654 | 0.6216 | 0.5975 | 0.7346 | 0.6483 | 0.6474 | 0.6731 | 0.6102 | 0.6280 | 0.6572 | 0.6201 |
| trafilatura-text | 0.5084 | 0.7416 | 0.6010 | 0.5762 | 0.7166 | 0.6319 | 0.6140 | 0.6534 | 0.5951 | 0.6162 | 0.6357 | 0.6025 |
| html2text-md | 0.4830 | 0.6093 | 0.6032 | 0.6637 | 0.6065 | 0.6877 | 0.5093 | 0.5967 | 0.6031 | 0.6165 | 0.6310 | 0.5927 |
| boilerpy3-text | 0.5009 | 0.6802 | 0.5361 | 0.4822 | 0.6630 | 0.5417 | 0.5672 | 0.5559 | 0.5311 | 0.5284 | 0.5661 | 0.4851 |
| newsplease | 0.4599 | 0.5631 | 0.5055 | 0.4752 | 0.6435 | 0.4688 | 0.5134 | 0.4816 | 0.5240 | 0.4541 | 0.5474 | 0.4892 |
| gne | 0.4150 | 0.6554 | 0.5287 | 0.4422 | 0.6664 | 0.5294 | 0.4669 | 0.5335 | 0.5081 | 0.5099 | 0.5393 | 0.4806 |
| justtext | 0.5003 | 0.5195 | 0.4805 | 0.4236 | 0.6053 | 0.4322 | 0.5200 | 0.4454 | 0.4888 | 0.4387 | 0.5201 | 0.4683 |
| boilerpy3-html | 0.3806 | 0.5962 | 0.4893 | 0.4802 | 0.5605 | 0.5173 | 0.4603 | 0.4998 | 0.4573 | 0.4627 | 0.5189 | 0.4421 |
| goose3 | 0.4256 | 0.4930 | 0.4312 | 0.3864 | 0.5748 | 0.3966 | 0.4815 | 0.4124 | 0.4661 | 0.4087 | 0.4785 | 0.3786 |
| readerlm | 0.1651 | 0.3002 | 0.2556 | 0.2045 | 0.3151 | 0.2485 | 0.2119 | 0.2357 | 0.2234 | 0.2113 | 0.2586 | 0.1899 |

Table 14: Detailed ROUGE-N F1 Scores across Topics (Part 2/2).

| Method | Health | Entertainment | Transportation | Hardware | Art & Design | Games | Fashion | Religion | Software Dev | Travel | Industrial | Adult |
|---|---|---|---|---|---|---|---|---|---|---|---|---|
| Dripper_fallback | 0.8806 | 0.8331 | 0.8320 | 0.8135 | 0.7774 | 0.7823 | 0.7526 | 0.8674 | 0.8420 | 0.8017 | 0.8386 | 0.7948 |
| Dripper | 0.8555 | 0.8144 | 0.8124 | 0.7832 | 0.7756 | 0.7509 | 0.7315 | 0.8357 | 0.8156 | 0.7908 | 0.8219 | 0.7927 |
| magic-html | 0.7776 | 0.6662 | 0.6716 | 0.6705 | 0.5899 | 0.6890 | 0.6061 | 0.7775 | 0.7299 | 0.6582 | 0.6930 | 0.6482 |
| readability | 0.7219 | 0.6465 | 0.5678 | 0.5964 | 0.5314 | 0.6206 | 0.4388 | 0.7286 | 0.6631 | 0.6444 | 0.6218 | 0.5961 |
| trafilatura-html | 0.7126 | 0.6314 | 0.5852 | 0.5818 | 0.5047 | 0.5971 | 0.5003 | 0.6762 | 0.5692 | 0.6133 | 0.5968 | 0.5980 |
| resiliparse | 0.7031 | 0.6025 | 0.6071 | 0.5590 | 0.5157 | 0.5841 | 0.5112 | 0.6684 | 0.6051 | 0.6276 | 0.5663 | 0.5685 |
| trafilatura-md | 0.6969 | 0.6212 | 0.5785 | 0.5628 | 0.5013 | 0.5853 | 0.4971 | 0.6665 | 0.5271 | 0.6023 | 0.5829 | 0.5930 |
| trafilatura-text | 0.6782 | 0.6046 | 0.5571 | 0.5469 | 0.4837 | 0.5738 | 0.4755 | 0.6465 | 0.5123 | 0.5823 | 0.5643 | 0.5763 |
| html2text-md | 0.6188 | 0.5826 | 0.5556 | 0.5254 | 0.4761 | 0.6154 | 0.3885 | 0.6758 | 0.7313 | 0.5124 | 0.5495 | 0.5514 |
| boilerpy3-text | 0.6423 | 0.5238 | 0.5515 | 0.4783 | 0.5037 | 0.4785 | 0.4175 | 0.6242 | 0.3926 | 0.5793 | 0.5297 | 0.5465 |
| newsplease | 0.5912 | 0.4901 | 0.5032 | 0.4410 | 0.4857 | 0.4316 | 0.3565 | 0.6161 | 0.4519 | 0.4845 | 0.4131 | 0.4749 |
| gne | 0.6194 | 0.5116 | 0.4756 | 0.4428 | 0.4366 | 0.4467 | 0.3772 | 0.5425 | 0.4698 | 0.4671 | 0.5052 | 0.5473 |
| justtext | 0.5685 | 0.4645 | 0.5030 | 0.4580 | 0.4644 | 0.4758 | 0.3988 | 0.5481 | 0.3159 | 0.5447 | 0.4004 | 0.4530 |
| boilerpy3-html | 0.5584 | 0.4624 | 0.4471 | 0.3716 | 0.4052 | 0.4428 | 0.3018 | 0.5619 | 0.4264 | 0.4516 | 0.4295 | 0.4986 |
| goose3 | 0.5192 | 0.4299 | 0.4599 | 0.3936 | 0.3853 | 0.3804 | 0.3594 | 0.5662 | 0.3011 | 0.4487 | 0.3846 | 0.4083 |
| readerlm | 0.2732 | 0.1949 | 0.2185 | 0.1930 | 0.1797 | 0.2047 | 0.1157 | 0.2638 | 0.1977 | 0.1964 | 0.1989 | 0.2366 |

Table 15: Detailed ROUGE-N F1 Scores across Formats (Part 1/2).

| Method | Comment Section | Structured Data | About (Org.) | About (Pers.) | Tutorial | Product Page | Content Listing | Customer Support | User Review | Spam / Ads | News (Org.) | Knowledge Article |
|---|---|---|---|---|---|---|---|---|---|---|---|---|
| Dripper_fallback | **0.7843** | **0.7310** | **0.8370** | **0.7547** | **0.9129** | **0.7971** | **0.6961** | **0.8392** | **0.8042** | **0.8178** | **0.9078** | **0.9047** |
| Dripper | 0.7415 | 0.7020 | 0.8283 | 0.7229 | 0.9018 | 0.7865 | 0.6646 | 0.8348 | 0.7868 | 0.8039 | 0.9000 | 0.8986 |
| magic-html | 0.4510 | 0.6045 | 0.7043 | 0.6254 | 0.8048 | 0.6425 | 0.5218 | 0.7598 | 0.5396 | 0.6960 | 0.8255 | 0.8142 |
| readability | 0.4429 | 0.4765 | 0.6429 | 0.5260 | 0.7666 | 0.5365 | 0.4500 | 0.6371 | 0.5104 | 0.6965 | 0.7723 | 0.7870 |
| trafilatura-html | 0.5362 | 0.4261 | 0.6637 | 0.5967 | 0.7301 | 0.5448 | 0.4888 | 0.6849 | 0.5837 | 0.6487 | 0.7410 | 0.7355 |
| trafilatura-md | 0.5364 | 0.4201 | 0.6404 | 0.5788 | 0.7122 | 0.5245 | 0.4608 | 0.6583 | 0.5855 | 0.6358 | 0.7387 | 0.7216 |
| resiliparse | 0.4799 | 0.4261 | 0.6177 | 0.5664 | 0.7260 | 0.5541 | 0.4932 | 0.6913 | 0.6074 | 0.5594 | 0.6908 | 0.7260 |
| trafilatura-text | 0.5311 | 0.3986 | 0.6296 | 0.5620 | 0.6895 | 0.5068 | 0.4486 | 0.6302 | 0.5787 | 0.6144 | 0.7127 | 0.7005 |
| html2text-md | 0.6010 | 0.6363 | 0.5113 | 0.4970 | 0.6975 | 0.4573 | 0.5982 | 0.6274 | 0.4946 | 0.5001 | 0.5491 | 0.6227 |
| boilerpy3-text | 0.4256 | 0.2503 | 0.6468 | 0.4808 | 0.6271 | 0.5128 | 0.2840 | 0.6163 | 0.5338 | 0.5295 | 0.6815 | 0.6333 |
| gne | 0.2889 | 0.2810 | 0.6057 | 0.4517 | 0.5841 | 0.4470 | 0.2974 | 0.5013 | 0.4403 | 0.5613 | 0.6649 | 0.6256 |
| newsplease | 0.3592 | 0.2387 | 0.5726 | 0.4586 | 0.6609 | 0.4437 | 0.3417 | 0.6157 | 0.5394 | 0.3120 | 0.5894 | 0.6461 |
| justtext | 0.5093 | 0.1470 | 0.6460 | 0.4728 | 0.6483 | 0.4604 | 0.2739 | 0.6219 | 0.6065 | 0.1823 | 0.5098 | 0.5447 |
| boilerpy3-html | 0.3933 | 0.2646 | 0.4876 | 0.4081 | 0.5856 | 0.3799 | 0.2572 | 0.5328 | 0.4197 | 0.4704 | 0.5436 | 0.5362 |
| goose3 | 0.2805 | 0.1509 | 0.5731 | 0.3824 | 0.6198 | 0.4131 | 0.2293 | 0.5034 | 0.4695 | 0.1697 | 0.5335 | 0.5632 |
| readerlm | 0.1540 | 0.1168 | 0.2597 | 0.1493 | 0.3050 | 0.1556 | 0.1259 | 0.2683 | 0.1886 | 0.2269 | 0.2945 | 0.2647 |

Table 16: Detailed ROUGE-N F1 Scores across Formats (Part 2/2).

| Method | Academic Writing | Personal Blog | Creative Writing | FAQ | Nonfiction Writing | Truncated | News Article | Audio Transcript | Legal Notices | Documentation | Listicle | Q&A Forum |
|---|---|---|---|---|---|---|---|---|---|---|---|---|
| Dripper_fallback | **0.8943** | **0.9086** | **0.8913** | **0.9280** | **0.9756** | 0.7097 | **0.9535** | **0.9555** | **0.9462** | **0.8769** | **0.9423** | **0.8303** |
| Dripper | 0.8444 | 0.8927 | 0.8097 | 0.8609 | 0.9641 | **0.7164** | 0.9466 | 0.9033 | 0.9192 | 0.8174 | 0.9251 | 0.8207 |
| magic-html | 0.8533 | 0.7698 | 0.8137 | 0.5514 | 0.9301 | 0.4891 | 0.8865 | 0.8881 | 0.9066 | 0.8082 | 0.8673 | 0.5470 |
| readability | 0.7765 | 0.7291 | 0.7896 | 0.6627 | 0.9179 | 0.4756 | 0.8693 | 0.8519 | 0.8916 | 0.7552 | 0.8033 | 0.6228 |
| trafilatura-html | 0.7933 | 0.7141 | 0.7516 | 0.6959 | 0.8700 | 0.4748 | 0.8330 | 0.8526 | 0.7982 | 0.6028 | 0.7814 | 0.5921 |
| trafilatura-md | 0.7781 | 0.7193 | 0.7252 | 0.7175 | 0.8634 | 0.4678 | 0.8268 | 0.8322 | 0.7780 | 0.5510 | 0.7772 | 0.5816 |
| resiliparse | 0.7767 | 0.7458 | 0.7129 | 0.7701 | 0.8894 | 0.4148 | 0.7675 | 0.8369 | 0.8034 | 0.6662 | 0.7737 | 0.7163 |
| trafilatura-text | 0.7628 | 0.7053 | 0.7050 | 0.6847 | 0.8484 | 0.4577 | 0.7973 | 0.8214 | 0.7589 | 0.5347 | 0.7521 | 0.5608 |
| html2text-md | 0.7734 | 0.6064 | 0.7053 | 0.6914 | 0.7577 | 0.3122 | 0.5856 | 0.8108 | 0.8701 | 0.8293 | 0.6909 | 0.4195 |
| boilerpy3-text | 0.6720 | 0.6399 | 0.6485 | 0.6150 | 0.8390 | 0.4478 | 0.7804 | 0.7502 | 0.7463 | 0.4333 | 0.6640 | 0.4872 |
| gne | 0.6584 | 0.5974 | 0.6908 | 0.4510 | 0.8065 | 0.4224 | 0.7529 | 0.7052 | 0.7880 | 0.4829 | 0.6907 | 0.4033 |
| newsplease | 0.6260 | 0.6915 | 0.5124 | 0.6744 | 0.8476 | 0.3524 | 0.6324 | 0.7799 | 0.7123 | 0.4373 | 0.7317 | 0.6525 |
| justtext | 0.6370 | 0.7155 | 0.4381 | 0.7084 | 0.8461 | 0.3432 | 0.5643 | 0.7810 | 0.6824 | 0.3567 | 0.7281 | 0.5653 |
| boilerpy3-html | 0.6422 | 0.5575 | 0.6371 | 0.6178 | 0.7643 | 0.3265 | 0.6710 | 0.7428 | 0.8026 | 0.5231 | 0.6212 | 0.3654 |
| goose3 | 0.5321 | 0.6184 | 0.3610 | 0.6698 | 0.8174 | 0.3836 | 0.6142 | 0.7118 | 0.5075 | 0.3190 | 0.6972 | 0.3404 |
| readerlm | 0.3292 | 0.2374 | 0.2802 | 0.2861 | 0.4097 | 0.1237 | 0.3553 | 0.3630 | 0.3996 | 0.2452 | 0.3138 | 0.1282 |

## A.13 USE OF LARGE LANGUAGE MODELS

A large language model is used as a writing assistant during the preparation of this manuscript. The primary use of the LLM is for improving grammar, clarity, and phrasing of the text. The LLM does not contribute to the core research ideas, experimental design, data analysis, or the formulation of our conclusions. The authors have reviewed and edited all text and take full responsibility for the final content of this paper.

```
f"""As a front-end engineering expert in HTML, your task is to analyze
   ↪   the given HTML structure and accurately classify elements with the
   ↪   {ITEM_ID_ATTR} attribute as either "main" (primary content) or
   ↪   "other" (supplementary content). Your goal is to precisely extract
   ↪   the primary content of the page, ensuring that only the most
   ↪   relevant information is labeled as "main" while excluding
   ↪   navigation, metadata, and other non-essential elements.
Guidelines for Classification:
Primary Content ("main")
Elements that constitute the core content of the page should be
   ↪   classified as "main". These typically include:
 For Articles, News, and Blogs:
The main text body of the article, blog post, or news content.
Images embedded within the main content that contribute to the article.
 For Forums & Discussion Threads:
The original post in the thread.
Replies and discussions that are part of the main conversation.
 For Q&A Websites:
The question itself posted by a user.
Answers to the question and replies to answers that contribute to the
   ↪   discussion.
 For Other Content-Based Pages:
Any rich text, paragraphs, or media that serve as the primary focus of
   ↪   the page.
Supplementary Content ("other")
Elements that do not contribute to the primary content but serve as
   ↪   navigation, metadata, or supporting information should be
   ↪   classified as "other". These include:
 Navigation & UI Elements:
Menus, sidebars, footers, breadcrumbs, and pagination links.
"Skip to content" links and accessibility-related text.
 Metadata & User Information:
Article titles, author names, timestamps, and view counts.
Like counts, vote counts, and other engagement metrics.
 Advertisements & Promotional Content:
Any section labeled as "Advertisement" or "Sponsored".
Social media sharing buttons, follow prompts, and external links.
 Related & Suggested Content:
"Read More", "Next Article", "Trending Topics", and similar sections.
Lists of related articles, tags, and additional recommendations.
Task Instructions:
You will be provided with a simplified HTML structure containing
   ↪   elements with an {ITEM_ID_ATTR} attribute. Your job is to analyze
   ↪   each element's function and determine whether it should be
   ↪   classified as "main" or "other".
Response Format:
Return a JSON object where each key is the {ITEM_ID_ATTR} value, and the
   ↪   corresponding value is either "main" or "other", as in the
   ↪   following example:
{{"1": "other","2": "main","3": "other"}}
Important Notes:
Do not include any explanations in the output, only return the JSON.
Ensure high accuracy by carefully distinguishing between primary content
   ↪   and supplementary content.
Err on the side of caution, if an element seems uncertain, classify it
   ↪   as "other" unless it clearly belongs to the main content.

Input HTML:
{html_str}

Output format should be a JSON-formatted string representing a
   ↪   dictionary where keys are item_id strings and values are either
   ↪   'main' or 'other'. Make sure to include ALL item_ids from the
   ↪   input HTML
"""
```

Figure 6: Prompt template for Main HTML classification.

