# OpenReview forum: "Dripper: Token-Efficient Main HTML Extraction with a Lightweight LM"
_ICLR.cc/2026/Conference — Submitted to ICLR 2026_

### Official Review · Reviewer_Gtgx · 2025-11-01

**Soundness:** 2
**Presentation:** 1
**Contribution:** 3
**Rating:** 2
**Confidence:** 3

**Summary:**

This paper proposes the Dripper framework, which addresses issues like context window limits, high inference costs, and hallucinations in web main content extraction. Dripper has three core parts: HTML simplification, semantic block classification, and controlled decoding. The authors also build MainWebBench dataset to evaluate model performance. A model fine-tuned on Qwen3-0.6B outperforming traditional methods and generative large models significantly on MainWebBench, with inference costs only 5.18% of the naive approach.

**Strengths:**

- The pipeline, HTML simplification + block classification + controlled decoding, directly solves efficiency and hallucination issues.
- Comprehensive experiments including 12 baseline methods and 2 datasets ensure the performance of Dripper. and public code, models, and datasets ensure reproducibility.
- MainWebBench offers community value on model evaluation, which is 7x larger than existing datasets, with complete annotations.
- The 0.6B-parameter model suits large-scale processing, meeting real-world needs.

**Weaknesses:**

- The presentation of figures, equations, and tables in this paper has inconsistencies affecting clarity. For example, the color scheme of Fig. 2 is confusing and lacks a corresponding description. There is a formatting mismatch between the description of Eq. (1) and the equation itself. Additionally, the captions for tables are overly simplistic, making it difficult to understand the tables.
- Lack of analysis on multilingual (e.g., Chinese) and domain generalization; MainWebBench has no domain-specific evaluation.
- No quantitative justification for key HTML simplification thresholds (e.g., 200-character truncation, line 203), with subjective settings affecting reproducibility.
- Lack of specific details for processing and model training.

**Questions:**

- What are the specific criteria for "splitting tables" in HTML simplification (line 198)? Are there experiments verifying how this rule affects extraction accuracy?
- How does the FSM in controlled decoding ensure no semantic block item_ids are missing (line 232)? Is there a fault-tolerance mechanism for failed state transitions?
- Can you add separate performance data for Chinese web pages in MainWebBench and compare accuracy differences between English and Chinese?
- In the fallback strategy, how do you determine Dripper has failed? What is the basis for choosing the failure threshold?

---

> ### Author Response · Authors · 2025-11-20
>
> $\textbf{Overall Response}$
>
> We thank the reviewer for the detailed comments. We have addressed all concerns by: (1) providing a detailed performance breakdown on Chinese and multilingual datasets to demonstrate domain generalization; (2) including ablation studies to justify our design choices; (3) providing details for processing and model training; and (4) clarifying the fault-tolerance mechanisms of our FSM and fallback strategies. We are grateful for the detailed suggestions with respect to the typos. The manuscript has been updated to reflect these additions.
>
>
> > $\textbf{Weakness 1 and Question 3: Multilingual evaluation}$
>
> > $\textbf{Response}$ :
>
> We appreciate the reviewer's interest in multilingual support. Constructing MainWebBench to approximate the linguistic diversity of Common Crawl is a core motivation of our work. **Our analysis confirms that the benchmark effectively reflects this diversity, ensuring a robust evaluation across different languages.**
>
> In our benchmark, 90% of the data is sampled from Common Crawl, which inherently contains web pages in various languages. **Among the 7,887 samples, 6,710 (85.09%) are English pages, 716 (9.1%) are Chinese, and 460 (5.83%) are in other languages.** We regrouped the samples by language and recalculated the average metrics for each method. The results are summarized in the table below:
>
> | Method           | Mode      | all.f1  | en.f1   | zh.f1   | other.f1 |
> |------------------|-----------|---------|---------|---------|----------|
> | **Dripper_fallback** | **HTML+MD**   | **0.8399**  | **0.8387**  | **0.8829**  | **0.7899**   |
> | **Dripper**          | **HTML+MD**   | **0.8182**  | **0.8165**  | **0.8663**  | **0.7694**   |
> | magic-html       | HTML+MD   | 0.7091  | 0.7054  | 0.7719  | 0.6653   |
> | readability      | HTML+MD   | 0.6491  | 0.6471  | 0.7108  | 0.5828   |
> | trafilatura      | HTML+MD   | 0.6358  | 0.6341  | 0.6902  | 0.5752   |
> | trafilatura      | Markdown  | 0.6237  | 0.6208  | 0.6847  | 0.5702   |
> | resiliparse      | Text      | 0.6233  | 0.6327  | 0.5662  | 0.5755   |
> | trafilatura      | Text      | 0.6049  | 0.6045  | 0.6414  | 0.5549   |
> | html2text        | Markdown  | 0.5977  | 0.6052  | 0.552   | 0.5593   |
> | boilerpy3        | Text      | 0.5413  | 0.5498  | 0.4824  | 0.5081   |
> | gne              | HTML+MD   | 0.5148  | 0.5132  | 0.5591  | 0.4704   |
> | newsplease       | Text      | 0.5012  | 0.5499  | 0.143   | 0.3479   |
> | justtext         | Text      | 0.477   | 0.5549  | 0.0067  | 0.0722   |
> | boilerpy3        | HTML+MD   | 0.4766  | 0.4825  | 0.4285  | 0.4657   |
> | goose3           | Text      | 0.4354  | 0.4899  | 0.0045  | 0.3113   |
> | readerlm         | Markdown  | 0.2264  | 0.226   | 0.2478  | 0.1987   |
>
> As observed, the performance ranking of the methods remains relatively consistent across different languages (with exceptions like newsplease, justtext, and goose3, which show significant performance drops on non-English pages). **Our method performs well across all languages**. This can be attributed firstly to the strong multilingual support inherent in our chosen base model, Qwen3-0.6B, and secondly to our **balanced sampling of training data** from Common Crawl, which carefully considered English, Chinese, and other languages.

---

> > ### Author Response · Authors · 2025-11-20
> >
> > > $\textbf{Weakness 2: Domain-specific evaluation}$
> >
> > > $\textbf{Response}$ :
> >
> > **MainWebBench is designed to cover a wide spectrum of specific domains, on which our method demonstrates exceptional robustness and generalizability.** To quantify this, we utilized the Topic and Format classifiers proposed in "Organize the Web: Constructing Domains Enhances Pre-Training Data Curation[1]" to categorize all 7,887 samples in the MainWebBench. Based on this classification, we calculated the ROUGE-N F1 scores for all methods across **24 distinct topics** (e.g., Science & Tech, Finance, Health) and **24 distinct formats** (e.g., News Article, Tutorial, Forum).
> >
> > Given the substantial volume of data generated (involving detailed scores across 48 sub-categories), we report only the **average rank** for each method in this response. The full, detailed results have been included **in the Appendix A.12**.
> >
> > | Tool | Topic Score | Topic Rank | Format Score | Format Rank |
> > | :--- | :---: | :---: | :---: | :---: |
> > | **Dripper_fallback** | 0.8583 | 1 | 0.8320 | 1 |
> > | **Dripper** | 0.8330 | 2 | 0.8102 | 2 |
> > | **magic-html** | 0.7209 | 3 | 0.6974 | 3 |
> > | **readability** | 0.6829 | 4 | 0.6342 | 4 |
> > | **trafilatura-html-md** | 0.6725 | 5 | 0.6259 | 5 |
> > | **resiliparse** | 0.6672 | 6 | 0.6122 | 7 |
> > | **trafilatura-md** | 0.6596 | 7 | 0.6144 | 6 |
> > | **trafilatura-text** | 0.6413 | 8 | 0.5956 | 8 |
> > | **html2text-md** | 0.6185 | 9 | 0.5827 | 9 |
> > | **boilerpy3-text** | 0.5811 | 10 | 0.5377 | 10 |
> > | **newsplease** | 0.5595 | 11 | 0.4944 | 12 |
> > | **gne** | 0.5499 | 12 | 0.5049 | 11 |
> > | **justtext** | 0.5412 | 13 | 0.4766 | 13 |
> > | **boilerpy3-html-md** | 0.5229 | 14 | 0.4676 | 14 |
> > | **goose3** | 0.4775 | 15 | 0.4321 | 15 |
> > | **readerlm** | 0.2492 | 16 | 0.2205 | 16 |
> >
> > As shown in the table, **Dripper_fallback** consistently secures the **1st place** ranking **across various topic and format categories**. On average, it outperforms the strongest existing baseline (magic-html) by **19%**, demonstrating the exceptional robustness of this strategy. The standalone **Dripper** model consistently ranks **second**. These results consistently demonstrate our model's **strong capabilities across diverse domains**.
> >
> > [1] Wettig A, Lo K, Min S, et al. Organize the Web: Constructing Domains Enhances Pre-Training Data Curation[J]. arXiv preprint arXiv:2502.10341, 2025.
> >
> > > $\textbf{Weakness 3: HTML simplification thresholds}$
> >
> > > $\textbf{Response}$ :
> >
> > **Our additional experiments verify that the truncation length of 200 is the optimal threshold, achieving the best performance balance**. To strictly evaluate the impact of text truncation length on model performance, we conducted the following ablation study. The results are summarized in the table below:
> >
> > | Truncation Length | F1 Score |
> > |-------------------|----------|
> > | 1                 | 51.86    |
> > | 10                | 75.46    |
> > | 50                | 81.32    |
> > | **200 (Ours)**        | **81.82**    |
> > | 500               | 81.67    |
> > | 1000              | 81.58    |
> > | 2000              | 81.55    |
> >
> > As the results indicate, in the most extreme scenario (max_length=1), the model fails to utilize intra-block semantic cues to determine the main content scope, resulting in poor performance (51.86). However, retaining even a minimal amount of text (max_length=10) yields a substantial performance boost. This confirms that our model indeed **relies on semantic understanding of the text content** to identify the main HTML boundaries.Notably, performance effectively converges when the truncation length reaches 50, approaching the peak score (81.32 vs. 81.82).
> >
> > Furthermore, extending the length beyond 200 **reveals a negligible impact on performance** (fluctuating within ~0.3%). This demonstrates that a truncation length of 200 captures sufficient semantic information for accurate classification while maintaining efficiency, confirming it as **a reasonable threshold**.

---

> > > ### Author Response · Authors · 2025-11-20
> > >
> > > > $\textbf{Weakness 4-1: Details for model training}$
> > >
> > > > $\textbf{Response}$ :
> > >
> > > We have added a detailed description of our experimental setup to Appendix A.11 in the revised PDF. The specific hyperparameters and training configurations are summarized in the table below:
> > >
> > > | Category               | Parameter                          | Value                              |
> > > |------------------------|------------------------------------|------------------------------------|
> > > | Model & Framework      | Base Model                         | Qwen3-0.6B                         |
> > > |                        | Fine-tuning Method                 | Full-parameter SFT |
> > > |                        | Training Framework                 | LLaMA-Factory[2]|
> > > | Training Dynamics      | Epochs                             | 4                                  |
> > > |                        | Global Batch Size                  | 128                                |
> > > |                        | Max Sequence Length                | 32,000                             |
> > > | Optimizer & Scheduler  | Learning Rate Scheduler            | Cosine                             |
> > > |                        | Peak Learning Rate                 | 1.00E-04                           |
> > > |                        | Warmup Ratio                       | 0.1                                |
> > > | Hardware & Efficiency  | Hardware                           | 32x NVIDIA A100 GPU                |
> > > |                        | Precision                          | BF16                               |
> > >
> > > We believe this addition fully clarifies our experimental procedure and resolves ambiguity. We appreciate the reviewer for helping us improve the clarity of our paper.
> > >
> > > [2] Zheng Y, Zhang R, Zhang J, et al. Llamafactory: Unified efficient fine-tuning of 100+ language models[J]. arXiv preprint arXiv:2403.13372, 2024.

---

> > > > ### Author Response · Authors · 2025-11-20
> > > >
> > > > > $\textbf{Weakness 4-2: Details for processing}$
> > > >
> > > >
> > > > > $\textbf{Response}$ :
> > > >
> > > > We appreciate the opportunity to elaborate on our HTML simplification algorithm, which is a cornerstone of the Dripper framework. While the main paper outlines the overall workflow and we have provided the full code for transparency, we are happy to describe the key steps of our pre-processing in greater detail here.
> > > >
> > > > The primary goal of our simplification algorithm is to **drastically reduce HTML token count while preserving the critical semantic and structural cues necessary for accurate content classification**. This is achieved through a multi-stage process applied to the raw HTML:
> > > >
> > > > **DOM Cleaning and Pruning**: We first parse the HTML and proactively remove entire subtrees known to be boilerplate. This includes tags such as **\<script>, \<style>, \<header>, \<footer>, and \<nav>**. Furthermore, we heuristically remove elements whose **class** or **id** attributes contain keywords like 'nav', 'footer', or 'header', or which have CSS styles indicating they are hidden (e.g., **display: none**).
> > > >
> > > > **Attribute Simplification**: To reduce noise and token overhead, we strip nearly all attributes from all elements. The only exceptions are the **class** and **id** attributes, which are often the most informative semantic markers in modern web design, and for **\<img>** tags, we also preserve the **src** (excluding large base64 data) and **alt** attributes.
> > > >
> > > > **Semantic Block Segmentation**: The core of our method involves converting the cleaned DOM tree into a linear sequence of semantic blocks. We perform a recursive traversal of the DOM, segmenting it at natural block-level boundaries. Our algorithm intelligently handles mixed content:
> > > >
> > > > 1. It identifies and preserves atomic block-level elements (e.g., a standalone paragraph or **div**).
> > > >
> > > > 2. It aggregates consecutive inline elements (e.g., **span**, links with text) and unwrapped text nodes into coherent blocks, wrapping them in a custom tag if necessary to maintain structure.
> > > >
> > > > 3. It makes special provisions for complex structures like tables and lists, ensuring they are treated as single, indivisible units where appropriate.
> > > >
> > > > **Content Truncation within Blocks**: To handle excessively long blocks (e.g., a massive list or a very long paragraph), we apply a conservative truncation strategy. We recursively traverse the block's content, limiting the total plain text to a predefined maximum length (e.g., 200 characters) while meticulously preserving the overall HTML tag structure. This ensures the model receives a representative sample of the content for classification without being overwhelmed by length.
> > > >
> > > > A critical innovation is the **parallel generation of Simplified HTML and Mapping HTML**. Both representations undergo identical block segmentation, ensuring a one-to-one correspondence between blocks. However, **Simplified HTML** used for model input undergoes the **full pruning and truncation process (steps 1-4)**. In contrast, **Mapping HTML**, used for final output reconstruction, undergoes **only the initial cleaning (step 1) and segmentation (step 3), preserving the original, un-truncated content**.
> > > >
> > > > Finally, we add a unique **_item_id** attribute to each block in both the Simplified and Mapping HTML. This allows the classification labels produced by Dripper-0.6B on the simplified sequence to be precisely mapped back to the rich, original content blocks for the final extraction.
> > > >
> > > > We hope this detailed description clarifies the inner workings of our pre-processing pipeline and demonstrates its role in enabling efficient and accurate main content extraction. We have made the code publicly available. **Welcome to clone it for testing and refer to the code for more details**.

---

> > > > > ### Author Response · Authors · 2025-11-20
> > > > >
> > > > > > $\textbf{Question 1: Criteria for splitting table}$
> > > > >
> > > > > > $\textbf{Response}$ :
> > > > >
> > > > > The specific criteria for "splitting tables" involve **a heuristic strategy** that classifies a table as a **Data Table** and **Layout Table**. In Common Crawl data, we observe that the \<table> tag serves two distinct and conflicting purposes:
> > > > >
> > > > > **a) Data Tables**: Used to present structured information (e.g., product specs, schedules). These often contain many homogeneous elements. For main content extraction, treating such tables as a single atomic block improves processing efficiency.
> > > > >
> > > > > **b) Layout Tables**: Developers utilize the \<table> tag as a container to control the visual layout of page elements. For instance, a webpage might structure its entire layout within a single table:
> > > > >
> > > > > - Row 1: Contains the Site Banner.
> > > > > - Row 2: Contains the Navigation Bar.
> > > > > - Row 3, Col 1: Encapsulates the Main Content (the target).
> > > > > - Row 3, Col 2: Displays Sidebar Advertisements.
> > > > > - Row 4: Lists Related Articles.
> > > > >
> > > > > In this scenario, tables must be "flattened" (decomposed), allowing the model to process each cell as an independent unit.
> > > > >
> > > > > In real-world scenarios, **both patterns are prevalent and often coexist**; for instance, a data table may be nested within a page that uses a table for global layout. Indiscriminately flattening all table elements yields the theoretical optimal performance, but this often results in **suboptimal efficiency** due to the increased number of blocks. Conversely, uniformly treating all tables as indivisible blocks **renders the model completely ineffective** on pages utilizing table-based layouts.
> > > > >
> > > > >
> > > > > To balance this, we employ a heuristic strategy. A table is classified as a Data Table if it meets any of the following structural criteria; otherwise, it is treated as a Layout Table :
> > > > > - Contains **\<caption>, \<th>, \<thead>, \<tfoot>, \<colgroup>, or \<col>** tags.
> > > > > - Has **summary**, **role="table"**, or **data-table** attributes.
> > > > > - Cells contain **headers** attributes.
> > > > >
> > > > > We compared three strategies on MainWebBench: (a) Auto (use the above strategy); (b) Data-only (treat all as atomic); and (c) Layout-only (flatten all).
> > > > >
> > > > > | Strategy       | ROUGE F1 | Mean Block Num |
> > > > > |----------------|----------|----------------|
> > > > > | **Auto (Ours)**    | **81.82**    | **73.01**          |
> > > > > | Data-only      | 78.18    | 58.6           |
> > > > > | Layout-only    | 81.86    | 93.6           |
> > > > >
> > > > > As shown in the table, the Data-only strategy yields the lowest F1, confirming that layout tables are prevalent and problematic if ignored. The Layout-only strategy achieves a marginally higher score than Auto (+0.04) but **incurs a 28% increase in average block count**. This significant increase in sequence length disproportionately raises inference cost for a negligible performance gain. This confirms that our Auto strategy effectively strikes **an optimal balance between extraction accuracy and processing efficiency**.

---

> > > > > > ### Author Response · Authors · 2025-11-20
> > > > > >
> > > > > > >  $\textbf{Question 2: Fault-tolerance mechanism}$
> > > > > >
> > > > > > >  $\textbf{Response}$ :
> > > > > >
> > > > > > Our controlled decoding mechanism provides a theoretical guarantee for **both the syntactic correctness of the JSON output and the completeness of the extracted content**.
> > > > > >
> > > > > > The FSM in our controlled decoding is designed as a deterministic finite state machine that guarantees the generation of a syntactically perfect JSON output structure. To ensure no semantic block item_ids are missing, the FSM employs **a strict sequential counting mechanism** that starts with item_id = 1 and increments sequentially after processing each semantic block. This ensures every block from 1 to max_count is processed in order without gaps.
> > > > > >
> > > > > > The state transitions are mathematically closed within Python's positive integer domain, providing complete coverage of all required JSON syntactic tokens. The FSM progresses through outer states **(Begin → Decide_Main_Other → End → EOS)** and for each block, navigates inner states **(WaitingForQuote → BuildingNumber → WaitingForValue → WaitingForCommaOrEnd)** to systematically construct the output.
> > > > > >
> > > > > > While the state transitions are theoretically guaranteed, we've implemented **robust error detection** through state validation at each transition point. If an unexpected token is encountered, indicating potential model deviation, the system immediately **raises a DripperLogitsError with detailed context**. This transfers control back to the caller, enabling trying **fallback to alternative extraction methods, error logging, or retry mechanisms**.
> > > > > >
> > > > > > Empirically, the FSM has demonstrated perfect reliability throughout our evaluation on the MainWebBench dataset, processing all 7,887 samples **without encountering any state transition failures**. The combination of mathematically sound state transitions and proactive exception handling ensures **both complete item_id sequences and robust error recovery when needed**. We invite the reviewer to **examine the complete implementation in our code repository** under the **TokenStateMachine class** for further technical details.
> > > > > >
> > > > > > > $\textbf{Question 4: Fallback mechanism}$
> > > > > >
> > > > > > > $\textbf{Response}$ :
> > > > > >
> > > > > > We have implemented a comprehensive multi-stage failure detection mechanism to guarantee the robustness and stability of the extraction process.
> > > > > >
> > > > > > Specifically, we define extraction failure as the inability to extract main HTML content from a web page. Our code implements multiple detection mechanisms for such scenarios, primarily including:
> > > > > >
> > > > > > 1.Preprocessing failures, or when the preprocessed input data exceeds the model's context window (using Qwen3-0.6B's context length of 32K as the threshold)
> > > > > >
> > > > > > 2.Model inference failures
> > > > > >
> > > > > > 3.Post-processing failures
> > > > > >
> > > > > > 4.Cases where all blocks are labeled as 'other' after post-processing (indicating no main content was identified)
> > > > > >
> > > > > > Upon encountering any of these situations, **the system automatically falls back to the backup procedure by invoking Trafilatura** to re-extract the main HTML content.

---

### Official Review · Reviewer_ouM2 · 2025-11-01

**Soundness:** 4
**Presentation:** 3
**Contribution:** 3
**Rating:** 8
**Confidence:** 4

**Summary:**

This paper tackles extraction of main body content from scraped HTML webpages, which is an important piece of data curation for training and prompting language models. Baselines in this space are usually heuristic tools like trafilatura (used in corpora like RefinedWeb) or newer LM based approaches like ReaderLM v2 (which is the most similar work to this one). These baselines suffer from the usual issues – heuristic based techniques aren’t accurate enough and LM based techniques aren’t efficient enough to use at scale. What this work does to improve on prior work is finetune a small Qwen model on this HTML extraction task and show that it achieves both extraction fidelity and efficiency, best of both worlds.

Of course, finetuning a Qwen model is straightforward, so let’s focus on what this paper has done that’s distinguishing from a simple finetuning. First, they finetune not just a model but also define a system around the model that approaches the HTML content extraction problem as a block sequence classification problem. A preprocessor identifies block regions in the HTML page using HTML tags, turns this into a sequence of blocks, and each block gets classified using the model into categories such as “main” or “other”. To ensure the model adheres to this scheme and produces correctly formed JSON output, they define a constrained decoding module imposed over the model’s inference.

They evaluate this system over both an established benchmark (MCEB) which they establish a state of the art, as well as introduce their own benchmark using diversely sampled Common Crawl webpages, for which they also establish state of the art results over baselines.

**Strengths:**

Very solid submission and I’m appreciative of work like this that goes the full effort to build something usable in the real world (not just hill climbing on others’ benchmarks, but curating a good benchmark themselves; not just training a model but designing a system that can use the model, with considerations of deployment efficiency). The results speak for themselves; impressive improvement over past work on an important but under-appreciated problem. This paper should be accepted.

I’ve seen work like this critiqued in the past for being too “engineering-heavy” or critique the modeling aspect for being too straightforward (finetuning Qwen). But preemptively, I want to say that I don’t buy those arguments as this work also shows the underappreciated side of diving into the details of a challenging niche problem and showing it’s not just about finetuning but about also careful approach to the problem to designing a system (e.g. the approach as sequence classification).

**Weaknesses:**

I think this paper should be accepted. That being said, here are some aspects that bothered me a bit while reading the submission. I would really appreciate if the authors can revise accordingly:

First, I think it’s important for the authors to provide a discussion of relation to prior work in block sequence classification. For clean content extraction of structured, layout-rich documents, this has been done before. For example, “VILA: Improving Structured Content Extraction from Scientific PDFs Using Visual Layout Groups (TACL 2021)”, “Form2Seq : A Framework for Higher-Order Form Structure Extraction (EMNLP 2020)”. I believe on web HTML documents, there is old work that also explicitly looks at HTML web elements as blocks for classification, such as “Web2Text: Deep Structured Boilerplate Removal (ECIR 2018)”. I didn’t do a comprehensive search; just a quick search on “block classification” or “sequence classification” over “layout rich” documents and found quite a few. Of course, none of these are done in the autoregressive language modeling era, so that’s why I still value this current submission, but I would like to see some reference back to this family of work (maybe using the extra space in camera ready to add a paragraph in related work about it.)

Second, I am appreciative of the ablation for the constrained decoding mechanism; it’s quite intuitive that there’s less need for it as we increase the amount of training data. I would like to see some discussion about how this relates to two pockets of work that aren’t referenced here.  First, constrained decoding methods for sequence classification (or any structured prediction) is well studied back in the era of CRFs; would like to see a bit of reference back to that, why they disappeared roughly around the BERT + LLM scaling-up era, and why it makes sense to bring them back today for this HTML content extraction task. I think this will help readers better appreciate why there is this component; otherwise, even if there is an ablation, it feels extraneous as long as one can scale training data. The second is, with any constrained decoding, there is always a penalty to inference speed. For a paper like this, which seems to be highly motivated by the practical applications of the system, it is important to put some numbers down about the inference speed cost (or lack thereof). For example, standard fast inference frameworks like VLLM or SGLANG can run Qwen family models really quickly and also support structured decoding as part of their framework; could one have implemented your FSM natively within those frameworks? Even in those frameworks, there is an inference speed tradeoff.  I don’t think need a major discussion here, but something would be nice.

Besides this, if you’re able to provide, I could use a bit more motivation on why it is important to curate your new benchmark vs rely on MCEB. As a reader, I could benefit from a really straightforward answer to the question: “In what cases would I want to evaluate on MainWebBench? In what cases would I want to evaluate on MCEB? In what cases, both?”  It’s not clear what the tradeoffs between these benchmarks are; whether this work is claiming MainWebBench should replace MCEB because of X, Y, Z properties, or if it is complementary in X, Y, Z ways?

**Questions:**

Do you happen to have baseline numbers for resilparse? It is actually the more popular tool nowadays compared to trafiliatura.

---

> ### Author Response · Authors · 2025-11-20
>
> $\textbf{Overall Response}$
>
> We sincerely appreciate the reviewer's insightful comments and constructive feedback. We address all concerns by: (1) contextualizing Dripper against related work and discussing the paradigm shift in constrained decoding; (2) providing inference performance tests and the Resiliparse comparison; and (3) clarifying the unique necessity of MainWebBench as a native, modern evaluation set compared to legacy aggregations like MCEB.
>
> > $\textbf{Weakness 1: Connection to Prior Work in Block Sequence Classification}$
>
> > $\textbf{Response}$ :
>
> Thank you for bringing these interesting works and perspectives to our attention! We have conducted a detailed analysis of these works specifically from the perspective of sequence classification. To facilitate a clear comparison, we have summarized the key differences in the Table below. We will include a detailed discussion of these works in the final manuscript if our work is accepted.
>
> First, we distinguish our approach from visual-layout models such as LayoutLM, VILA, and Form2Seq. These models fundamentally rely on rendering information—utilizing 2D spatial coordinates or "visual layout groups" to aggregate text lines. While effective for PDFs or forms, this paradigm is ill-suited for large-scale web crawling. In our context, pages are typically archived as raw HTML text, lacking the external CSS, images, or rendering engines required to reconstruct the visual layout. Consequently, methods dependent on rendering or visual grouping are often inapplicable to raw crawl data.
>
> Secondly, we acknowledge the foundational contributions of pre-LLM HTML extraction works. Some of these works explicitly treated HTML extraction as a sequence labeling task. For example, Web2Text utilized DOM leaf nodes as atomic units and employed a Hidden Markov Model based on CNN potentials to perform sequence labeling. Similarly, BoilerNet utilized BiLSTMs to perform sequence labeling on DOM leaf nodes via automatic feature learning. Other works, like MarkupLM and WebFormer, modified model architectures (by adding XPath embeddings or graph attention) to explicitly capture tree topology for understanding web content. However, because these architectures were specialized modifications of pre-BERT, early BERT, or CNN/RNN baselines, they lack the massive pre-training on general world knowledge and refined post-training that characterize modern Large Language Models. As a result, they often struggle to generalize across the diverse semantics and noise of the modern open web.
>
> In contrast, our work does not modify the internal model structure of Qwen3 for HTML. Instead, we strategically design the input processing (via Simplification) and the output structure (via Constrained Decoding). This approach is designed to maximize the utility of advancements in the LLM community regarding open-source base models, inference acceleration, and hardware utilization. This makes the proposed content extraction tool significantly more practical and allows it to continuously incorporate future community advancements.
>
> Finally, ReaderLM-v2 represents another technical route using general generative language models, aligning closely with the "prompt in, response out" paradigm. However, this approach forces the model to ingest raw, noisy, and very long HTML to freely generate content. For Small Language Models, this free-form generation poses significant risks of hallucination in both content and format. The return on investment for solving these issues via pre-training and post-training is low. Therefore, at the current stage of development, directly using language models to generate main content may not be the optimal choice.
>
> In summary, the key differences between Dripper and prior works are:
>
> 1. Unlike works from the pre-decoder-only LM era, which primarily modified model architectures to adapt to HTML characteristics, we primarily adapt to HTML documents by designing the structure of input and output data.
>
> 2. Compared to concurrent works based on general Decoder-Only LMs, we leverage HTML structure to simplify input and output, avoiding the hallucination issues caused by generating rich content and formats.

---

> > ### Author Response · Authors · 2025-11-20
> >
> > | Work              | Domain     | Minimal Data Unit       | Core Model Architecture          | Structure Representation Mechanism                                                                                                                                                                                                                            |
> > |--------|------------|-------------------------|----------------------------------|---------------------------------------------------------------------------------------------------------------------------------------------------------------------------------------------------------------------------------------------------------------|
> > | LayoutLM[1]| PDF/Doc    | Token                   | Encoder-Only (Transformer)       | 2D Geometric Embeddings (Added to input layer; based on token bounding box coordinates)                                                                                                                                                                      |
> > | VILA[2]     | PDF/Doc    | Token                   | Encoder-Only (Transformer)       | Semantic-Layout Grouping (Uses special tokens to delimit visually grouped text blocks/lines)                                                                                                                                                                 |
> > | Form2Seq[3] | Forms (PDF)| Text Block / Widget     | Seq2Seq (BiLSTM Encoder + LSTM Decoder) | Spatial Features + Bahdanau Attention (Element coordinates used as input features)                                                                                                                                               |
> > | MarkupLM[4] | HTML       | Token                   | Encoder-Only (BERT)              | Symbolic-Hierarchical Embeddings (XPath path from root to token added to input layer)                                                                                                                                                                         |
> > | Web2Text[5]| HTML       | Text Block              | CNN / HMM (Hybrid)               | Handcrafted DOM-based Features (128 structural/text features refined by a CNN)                                                                                                                                                                               |
> > | BoilerNet[6] | HTML       | DOM Leaf Node           | Encoder-Only (BiLSTM)            | Automatic Feature Learning (Vector encoding DOM tag path (ancestors) + words in the node)                                                                                                                                                                     |
> > | WebFormer[7] | HTML       | DOM Node + Text Token   | GNN / Transformer (Hybrid)       | Graph Attention mechanism operating over DOM tree (parent, child, sibling relations)                                                                                                                                            |
> > | ReaderLM-v2[8] | HTML     | Raw HTML Sequence       | Decoder-Only                     | Implicit-Generative Learning (Learned implicitly by generating structured output via SFT & DPO)                                                                                                                                                    |
> > | Dripper              | HTML       | Semantic Block          | Decoder-Only                     | Block Sequence Classification (Simplify HTML to reduce token; Explicit FSM constraints on output structure)                                                                                                                           |

---

> > > ### Author Response · Authors · 2025-11-20
> > >
> > > [1] Xu Y, Li M, Cui L, et al. Layoutlm: Pre-training of text and layout for document image understanding[C]//Proceedings of the 26th ACM SIGKDD international conference on knowledge discovery & data mining. 2020: 1192-1200.
> > >
> > > [2] Shen, Zejiang, et al. "VILA: Improving structured content extraction from scientific PDFs using visual layout groups." Transactions of the Association for Computational Linguistics 10 (2022): 376-392.
> > >
> > > [3] Aggarwal M, Gupta H, Sarkar M, et al. Form2Seq: A framework for higher-order form structure extraction[C]//Proceedings of the 2020 Conference on Empirical Methods in Natural Language Processing (EMNLP). 2020: 3830-3840.
> > >
> > > [4] Li, Junlong, et al. "MarkupLM: Pre-training of text and markup language for visually rich document understanding." Proceedings of the 60th Annual Meeting of the Association for Computational Linguistics (Volume 1: Long Papers). 2022.
> > >
> > > [5] Vogels, Thijs, Octavian-Eugen Ganea, and Carsten Eickhoff. "Web2text: Deep structured boilerplate removal." European Conference on Information Retrieval. Cham: Springer International Publishing, 2018.
> > >
> > > [6] Leonhardt, Jurek, Avishek Anand, and Megha Khosla. "Boilerplate removal using a neural sequence labeling model." Companion Proceedings of the Web Conference 2020. 2020.
> > >
> > > [7] Wang, Qifan, et al. "Webformer: The web-page transformer for structure information extraction." Proceedings of the ACM Web Conference 2022. 2022.
> > >
> > > [8] Wang, Feng, et al. "Readerlm-v2: Small language model for HTML to markdown and JSON." arXiv preprint arXiv:2503.01151 (2025).

---

> > > > ### Author Response · Authors · 2025-11-20
> > > >
> > > > > $\textbf{Weakness 2-1: Historical context of structured prediction}$
> > > >
> > > > > $\textbf{Response}$ :
> > > >
> > > > We agree that explicit constrained decoding (such as CRFs) faded during the BERT era. However, we argue that its reintroduction in the LLM era is driven by a fundamental paradigm shift from discriminative classification to generative modeling.
> > > >
> > > > **Why it Vanished**: In the BiLSTM-CRF era, constrained by limited training data and parameter scale, models struggled to capture long-range dependencies independently. Consequently, they relied on CRF layers to introduce explicit global rules via transition matrices to correct predictions that were "locally optimal but globally invalid." However, with the advent of powerful Transformer encoders like BERT, self-attention mechanisms effectively internalized these dependencies into contextual embeddings. Large-scale pre-training further endowed models with robust representation capabilities. When models became sufficiently powerful, simple token-wise classification (Softmax) could implicitly handle most dependencies. At that point, the marginal performance gains from CRFs were insufficient to justify their massive computational overhead, leading to their gradual abandonment.
> > > >
> > > > **Why it Returned**: In the LLM era, decoder-only models have demonstrated astonishing semantic understanding, making them excellent choices for complex structured tasks (such as tool calling and information extraction). However, generative models are inherently probabilistic; even the most powerful models cannot entirely eliminate the uncertainty (hallucinations) introduced by sampling, a problem that is even more pronounced in smaller models. Therefore, constrained decoding has been reintroduced as a low-cost "hard guardrail." It allows us to leverage the LLM's powerful semantic reasoning to tackle complex tasks while—at the cost of constrained decoding—eliminating the instability caused by its probabilistic nature, thus effectively solving the task.
> > > >
> > > > **Rationale in HTML Extraction**: Reintroducing constraints in HTML extraction is both reasonable and necessary.
> > > >
> > > > First, unlike the "soft" probabilistic constraints of CRFs (which merely signal low transition probabilities), our method applies "hard" symbolic constraints via a logits processor. This ensures the output is syntactically perfect JSON, which is critical for automated downstream pipelines that have zero tolerance for parsing failures.
> > > >
> > > > Second, as described in our methodology, we reframe the extraction task as a "sequential chunk classification" task. The constraint mechanism effectively forces the generative model to behave like a deterministic classifier, restricting the vocabulary to only "main" or "other" at key decision points. This eliminates the risk of generation errors and leverages the LLM's semantic understanding while avoiding its generative instability.
> > > >
> > > > Our ablation studies support this, showing that this structural prior is particularly valuable in low-data scenarios. It effectively guides the model before it has fully learned the format from the data, enabling us to achieve SOTA-level stability with fewer parameters and less data, rendering a 0.6B model production-ready. Crucially, this design ensures our architecture remains effective for future, smaller models; it guarantees output stability and maximizes performance even without millions of training samples.

---

> > > > > ### Author Response · Authors · 2025-11-20
> > > > >
> > > > > > $\textbf{Weakness 2-2: Inference performance}$
> > > > >
> > > > >
> > > > > > $\textbf{Response}$ :
> > > > >
> > > > > We sincerely thank the reviewer for their attention to the inference performance of our work. To address this, we evaluated the average execution time with and without the Finite State Machine (FSM) on an Nvidia H200. The testing protocol was as follows: we randomly selected 20 samples from the test set for warm-up, followed by testing on a separate set of 50 samples. Inference was performed serially (batch_size=1) on a single-process, single-GPU VLLM instance. We recorded both the mean and standard deviation of the execution time. We compared checkpoints trained on 2K data versus 870K data, under both FSM-constrained and unconstrained settings. The results are presented below:
> > > > >
> > > > > | Condition      | Mean (s) | Std (s) |
> > > > > |----------------|----------|---------|
> > > > > | 2K / w FSM     | 2.3      | 4.76    |
> > > > > | 2K / wo FSM    | 8.24     | 17.61   |
> > > > > | 870K / w FSM   | 2.29     | 4.66    |
> > > > > | 870K / wo FSM  | 1.89     | 3.99    |
> > > > >
> > > > > As shown in the table, for the fully trained model (870k), disabling the FSM reduces inference time by approximately 17% relative to the FSM-constrained setting. In high-throughput scenarios, such as batch processing large-scale web crawls like Common Crawl, this performance gain is significant for cost reduction. However, in interactive scenarios like Web Agents, given the latency inherent in network communication and agent reasoning, this slight increase in inference latency remains within acceptable limits.
> > > > >
> > > > > Conversely, with limited synthetic data (2k), the inference time with FSM remains consistent with the fully trained model. This is because the state machine strictly constrains the output format, effectively decoupling inference time from the model's internal capability. Without the FSM, however, the average inference time increases significantly. We observed that in certain cases, the model **entered infinite loops**, continuously generating incrementing keys until hitting the context length limit. This uncontrolled generation severely hampers both inference efficiency and result accuracy.
> > > > >
> > > > > Based on these results, we draw the following conclusions:
> > > > >
> > > > > **For potential smaller base models or scenarios with insufficient training data**, constrained decoding is essential to mitigate the efficiency waste and accuracy loss caused by format hallucinations.
> > > > >
> > > > > **For our current configuration (0.6B model + 870K synthetic samples)**, constrained decoding introduces a ~17% performance penalty. Therefore, we recommend prioritizing unconstrained decoding for this specific setup, utilizing a fallback strategy to handle sporadic bad cases.
> > > > >
> > > > > We thank the reviewer again for this question. It provided us the opportunity to rigorously re-examine and measure the efficiency/performance trade-offs of these techniques under different conditions. This analysis not only brings key improvements to the practical deployment of our work but also clarifies the optimal scope for applying constrained decoding.

---

> > > > > > ### Author Response · Authors · 2025-11-20
> > > > > >
> > > > > > > $\textbf{Weakness 3: Motivation for MainWebBench}$
> > > > > >
> > > > > > > $\textbf{Response}$ :
> > > > > >
> > > > > > First, a primary motivation for creating MainWebBench was to enable a more precise and fine-grained evaluation of HTML main content extraction. As illustrated by the annotation tool in Appendix A.3, our ground truth is meticulously annotated **directly at the HTML tag level**. This tag-level annotation not only effectively measures the extractor's ability to identify the main content but also provides a unified foundation for diverse post-processing formats. Whether the desired final output is plain text or Markdown (as discussed in the paper), our benchmark provides a clear "source of truth" to evaluate the quality of upstream extraction.
> > > > > >
> > > > > > Secondly, we address the limitations inherent in existing data. MCEB is not a native benchmark but rather **an aggregation compiled from existing, older datasets**, where each sub-dataset is relatively **small** in scale. Fundamentally, this collection is limited by its **heterogeneous origins**: the constituent datasets were constructed for varying purposes and suffer from inconsistent annotation standards. They also contain **data quality issues** such as file encoding errors or script injections; we followed the methodology from a relevant survey [1] to address these issues during aggregation. Furthermore, we found that this collection offers **insufficient coverage** of the defining characteristics of the modern internet: specifically, highly diverse, multilingual, and long-tail data. The datasets within MCEB are predominantly **English-centric** and **do not originate from Common Crawl**—the massive web corpus containing petabytes of data that serves as the primary training foundation for many modern LLMs.
> > > > > >
> > > > > > MainWebBench was constructed specifically to bridge these gaps. Prior to our work, there was no single, large-scale, multi-dimensional benchmark designed specifically for modern main content extraction. Our benchmark is the result of **a unified, rigorous annotation process involving multiple rounds of human quality assurance** to ensure high accuracy and consistency. It explicitly covers **a broader range of data dimensions**—including metadata for language, style (e.g., conversational), quantitative difficulty levels, and rich content tags (e.g., tables, code, equations)—as detailed in the appendix on benchmark construction. Furthermore, the benchmark is **actively maintained**.
> > > > > >
> > > > > > We believe our newly annotated benchmark offers significant advantages in terms of quality, scale, and metadata richness. We included MCEB in our experiments to ensure a fair and comprehensive comparison with existing work that relies on it, further demonstrating the superior extraction performance of our model. We advocate for future research to utilize this dataset for evaluation.
> > > > > >
> > > > > > [1] Bevendorff, Janek, et al. "An empirical comparison of web content extraction algorithms." Proceedings of the 46th International ACM SIGIR Conference on Research and Development in Information Retrieval. 2023.

---

> > > > > > > ### Author Response · Authors · 2025-11-20
> > > > > > >
> > > > > > > >  $\textbf{Question 1: Resiliparse}$
> > > > > > >
> > > > > > >
> > > > > > > >  $\textbf{Response}$ :
> > > > > > >
> > > > > > > We have conducted experiments using Resiliparse on MainWebBench, and the comparison with other major baselines is presented in the table below:
> > > > > > >
> > > > > > > | Method           | Mode      | ROUGE-N F1 | ROUGE-N Precision | ROUGE-N Recall |
> > > > > > > |------------------|-----------|------------|-------------------|----------------|
> > > > > > > | **Dripper\_fallback** | **HTML+MD**   | **0.8399**     | **0.8761**            | **0.8587**         |
> > > > > > > | **Dripper**          | **HTML+MD**   | **0.8182**     | **0.851**             | **0.838**          |
> > > > > > > | magic-html       | HTML+MD   | 0.7091     | 0.7873            | 0.7169         |
> > > > > > > | readability      | HTML+MD   | 0.6491     | 0.7542            | 0.6407         |
> > > > > > > | trafilatura      | HTML+MD   | 0.6358     | 0.7262            | 0.6202         |
> > > > > > > | trafilatura      | Markdown  | 0.6237     | 0.699             | 0.6138         |
> > > > > > > | **resiliparse**      | **Text**      | **0.6233**     | **0.6103**            | **0.7058**         |
> > > > > > > | trafilatura      | Text      | 0.6049     | 0.6884            | 0.5877         |
> > > > > > > | html2text        | Markdown  | 0.5977     | 0.4877            | 0.9759         |
> > > > > > > | boilerpy3        | Text      | 0.5413     | 0.7105            | 0.5026         |
> > > > > > > | gne              | HTML+MD   | 0.5148     | 0.7363            | 0.4883         |
> > > > > > > | newsplease       | Text      | 0.5012     | 0.6123            | 0.4644         |
> > > > > > > | justtext         | Text      | 0.477      | 0.5567            | 0.4581         |
> > > > > > > | boilerpy3        | HTML+MD   | 0.4766     | 0.4781            | 0.5569         |
> > > > > > > | goose3           | Text      | 0.4354     | 0.5971            | 0.3917         |
> > > > > > > | readerlm         | Markdown  | 0.2264     | 0.2081            | 0.4065         |
> > > > > > >
> > > > > > >
> > > > > > > As the results indicate, Resiliparse achieves a relatively high recall (0.7058); however, this comes at the expense of lower precision (0.6103), suggesting a tendency to retain significant boilerplate noise alongside the main content. In terms of the overall F1 score, Resiliparse (0.6233) performs comparably to Trafilatura-Markdown (0.6237) but remains significantly behind Dripper (0.8182) and Dripper_fallback (0.8399). This highlights the limitation of heuristic rules in distinguishing subtle boilerplate from main text, whereas Dripper successfully achieves both high recall and high precision simultaneously. We have incorporated the results into the baseline comparison table in the revised PDF.

---

### Official Review · Reviewer_3Nd5 · 2025-11-03

**Soundness:** 3
**Presentation:** 3
**Contribution:** 2
**Rating:** 6
**Confidence:** 5

**Summary:**

This paper introduces Dripper, a framework for  HTML boilerplate removal; Dripper leverages a Qwen 3 0.6B finetune for efficent extraction. Dripper operates on a simplified version of a web page, and employs constraint decoding to ensure more faithful output. Alongside Dripper, this work introduces MainWebBench, a  benchmark of 7,800 human-annotated pages.

**Strengths:**

- Efficient approach to improve context extraction from HTML pages.
- The approach is generally well motivated.
- Paper introduces a more comprehensive benchmark for evaluating HTML extraction.

**Weaknesses:**

- The described constrain decoding mechanism is framed as a contribution; however, constraint decoding via a grammar is a built in feature in major inference, so this aspect of the pipeline, while sensible, it fairly straightforward.
  - Futher, Seciotn 5.4 shows that the mechanism has limited impact once enough supervision is provided.
- The proposed benchmark is comprised of 90% randomly sampled websites, and 10% head distribution websites. It would have been useful to get more statistics about benchmark composition, including website locale, type of website, statistics of domains in common crawl graph and so on.
  - Given that the task is sequence classification, it would have been more appropiate to measure precision and recall per block rather than ROUGE.
- There are a couple baselines missing; for rule-based toolkits, [resiliparse](https://resiliparse.chatnoir.eu/en/stable/) is often used due to its better recall (e.g., [DCLM](https://arxiv.org/abs/2406.11794), [OLMo 2](https://arxiv.org/abs/2501.00656)); to ground the task, it would have also been useful to see performance of frontier language models on the same input Dipper receives.

**Questions:**

## Questions:
- It is unclear what the role of Mapping HTML is. If Simplified HTML is in order, and Dripper does sequence classification, then what's the need to keep a separate representation?

## Minor typos and feedback
- All citations are missing a space. should be "C4 (Raffel et al., 2020)", not "C4(Raffel et al., 2020)"
- `\cite` / `\citet` is used instead of `\citep` in several places (e.g. 044)
- The paper frames the approach as sequence tagging, but, based on the backbone used (Qwen), the task is better described as text generation.

---

> ### Author Response · Authors · 2025-11-20
>
> $\textbf{Overall Response}$
>
> We sincerely appreciate the reviewer's insightful comments. We address all concerns by: (1) clarifying key technical designs (Constrained Decoding, Mapping HTML) and terminology choices; (2) justifying the use of ROUGE metrics; and (3) demonstrating superiority over Resiliparse and Frontier LLMs with detailed benchmark statistics.The manuscript has been updated accordingly.
>
> > $\textbf{Weakness 1: Constrained Decoding}$
>
>
> > $\textbf{Response：}$
>
> While we acknowledge that constrained decoding is an established technique in general text generation, we wish to emphasize that within the Dripper framework, it functions as a decisive enabler rather than a mere supplementary feature, empowering a lightweight 0.6B model to handle complex extraction tasks.
>
> First, Small Language Models are inherently prone to **hallucination**. In web extraction tasks, this intrinsic instability is amplified by **long sequence length**, where the probability of maintaining syntactic integrity naturally diminishes, significantly increasing the risk of 'format collapse'. In response, our mechanism effectively mitigates this by reframing the open-ended generation task into a deterministic classification process. By constraining the vocabulary to strictly 'main' or 'other' at critical decision points, we **eliminate the risk of generation errors while leveraging the LM's semantic understanding capabilities**, thereby bypassing its generative instability.
>
> Furthermore, output determinism is essential for **large-scale extraction pipelines**. As demonstrated in our ablation studies, this structural prior proves particularly valuable in **low-data regimes**, effectively guiding the model before it has fully internalized the format from the data. While the performance gap narrows with increased training data, this mechanism enables us to achieve **SOTA-level stability with significantly fewer parameters and less data**, rendering the 0.6B model **production-ready**. Crucially, this design ensures our architecture remains robust for **even smaller future models**; it guarantees output stability and maximizes performance even in the absence of millions of training samples.

---

> > ### Author Response · Authors · 2025-11-20
> >
> > > $\textbf{Weakness 2: MainWebBench Details}$
> >
> > > $\textbf{Response：}$
> >
> > We appreciate the opportunity to elaborate on the details of MainWebBench. The MainWebBench dataset ($\text{N=7,887}$) features **high diversity**, covering **5,945 unique domains and 150 distinct Top-Level Domains (TLDs)**, with a hybrid sampling strategy **capturing both the web's "long-tail" and popular sites**. Its composition includes **various subcategories** (e.g., News, Product Page, Forum) and **46 different languages** (85.09% English, 9.08% Chinese), confirming its broad spectrum of global regions and website types.
> >
> > Regarding **diversity**, the dataset covers **5,945 unique domains**, confirming that the data is not dominated by a few sources but possesses a high degree of diversity. It spans **150 distinct Top-Level Domains (TLDs)**, indicating that the benchmark covers a broad spectrum of global regions and website categories. We present a partial list of domains and the TLD distribution below (Tables 1 & 2).
> >
> > #### Table 1: Partial Domain Statistics (Top 10 Sorted by Sample Count)
> > | Domain                     | Sample Count | Percentage | Top Language | Top Style          | Top Level | Has Table | Has Code | Has Equation |
> > |----------------------------|--------------|------------|--------------|--------------------|-----------|-----------|----------|--------------|
> > | aniruddhadeb.com           | 39           | 0.49%      | en           | Article            | simple    | 1         | 9        | 36           |
> > | politics.stackexchange.com | 30           | 0.38%      | en           | Forum/Comment      | mid       | 0         | 0        | 0            |
> > | www.ask.com                | 29           | 0.37%      | en           | Article            | simple    | 1         | 0        | 3            |
> > | en.wikipedia.org           | 27           | 0.34%      | en           | Article            | hard      | 20        | 1        | 0            |
> > | www.china.org.cn           | 23           | 0.29%      | en           | Article            | simple    | 21        | 0        | 0            |
> > | money.cnn.com              | 22           | 0.28%      | en           | Article            | hard      | 18        | 0        | 7            |
> > | data.epo.org               | 21           | 0.27%      | en           | Article            | simple    | 21        | 0        | 0            |
> > | m.weibo.cn                 | 19           | 0.24%      | zh           | Forum/Comment      | simple    | 0         | 0        | 0            |
> > | spanish.china.org.cn       | 15           | 0.19%      | es           | Article            | simple    | 14        | 0        | 0            |
> > | china.org.cn               | 14           | 0.18%      | en           | Article            | mid       | 13        | 0        | 0            |
> >
> > #### Table 2: Partial Top-Level Domain (TLD) Distribution (Top 10)
> > | Metric      | com    | org    | cn     | net    | uk     | edu    | de     | au     | ru     | gov     |
> > | :---------- | :----- | :----- | :----- | :----- | :----- | :----- | :----- | :----- | :----- | :------ |
> > | **Count**   | 4550   | 816    | 459    | 318    | 235    | 180    | 101    | 94     | 69     | 59      |
> > | **Percentage** | 57.69% | 10.35% | 5.82%  | 4.03%  | 2.98%  | 2.28%  | 1.28%  | 1.19%  | 0.87%  | 0.75%    |
> >
> >
> > Regarding **web page subcategory distribution**, we utilized **GPT-5** to classify the semantic type of every page in the benchmark. As shown in Table 3, the dataset covers **a diverse range of page layouts**.
> >
> > #### Table 3: Web Page Subcategory Distribution (Top 10)
> > | Metric      | News  | Other article | Navigation listing | Product page | Blog  | Multiple data article | Forum  | Tutorial | Article with comment section | Other  |
> > | :---------- | :---- | :------------ | :----------------- | :----------- | :---- | :-------------------- | :----- | :------- | :--------------------------- | :----- |
> > | **Count**   | 1,605 | 1,506         | 1,046              | 1,022        | 671   | 647                   | 487    | 305      | 228                          | 165    |
> >
> > Regarding **language distribution**, the dataset includes web pages in **46 different languages**. We present partial language statistics in Table 4.
> >
> > #### Table 4: Partial Language Distribution (Top 10)
> > | Metric      | English | Chinese | Spanish | German | Japanese | Russian | French | Italian | Korean | Portuguese |
> > | :---------- | :------ | :------ | :------ | :----- | :------- | :------ | :----- | :------ | :----- | :--------- |
> > | **Count**   | 6711    | 716     | 61      | 51     | 48       | 45      | 36     | 22      | 20     | 17         |
> > | **Percentage** | 85.09%  | 9.08%    | 0.77%    | 0.65%   | 0.61%     | 0.57%    | 0.46%   | 0.28%    | 0.25%   | 0.22%       |
> >
> >
> > **The full data available on HuggingFace [1]**. We have added the above content to Appendix A.6 in the revised version.
> >
> > [1] https://huggingface.co/anonymous-s2wrvq/Dripper

---

> > > ### Author Response · Authors · 2025-11-20
> > >
> > > > $\textbf{Weakness 3:  Evaluation Metrics}$
> > >
> > > > $\textbf{Response：}$
> > >
> > > In fact, we **closely monitored these metrics (Precision, Recall, and F1) throughout our model development process** to assess internal classification performance. We provide these results in the table below:
> > >
> > > | Data Size | Block-level Precision | Block-level Recall | Block-level F1 | ROUGE-N F1 |
> > > |-----------|-----------------------|--------------------|----------------|------------|
> > > | 2k        | 0.877                 | 0.781              | 0.756          | 0.770      |
> > > | 5k        | 0.875                 | 0.81               | 0.781          | 0.796      |
> > > | 10k       | 0.888                 | 0.823              | 0.800          | 0.811      |
> > > | 100k      | 0.900                 | 0.838              | 0.821          | 0.829      |
> > > | 870k      | 0.898                 | 0.843              | 0.826          | 0.834      |
> > >
> > > While the data confirms our model's strong classification capability, we deliberately **chose ROUGE-N as our primary reporting metric** for three key reasons.
> > >
> > > First, our pre-processing creates blocks with **significant content length variance**. A block can range from a single boilerplate word to a 2,000-word main article. Standard classification metrics treat all blocks equally, meaning a model could achieve a high F1 score by correctly classifying hundreds of tiny boilerplate blocks while missing the single, massive main content block. This would yield a high classification score but a completely failed extraction.
> > >
> > > Secondly, ROUGE-N better **aligns with the end-user's objective**, which is to obtain the complete main text. By measuring the overlap between the extracted text and the ground truth, ROUGE implicitly weights blocks by their information content, ensuring that the metric reflects the actual utility of the output.
> > >
> > > Finally, prioritizing ROUGE-N ensures **consistency** with established benchmarks in the web extraction literature, where ROUGE-L or ROUGE-N are the standard metrics for comparison. We have included the detailed block-level analysis in the Appendix of the revised revision to provide a more comprehensive view of the model's behavior.

---

> > > > ### Author Response · Authors · 2025-11-20
> > > >
> > > > > $\textbf{Weakness 4:  Resiliparse and Frontier LLMs}$
> > > >
> > > > > $\textbf{Response：}$
> > > >
> > > > We conducted additional experiments on MainWebBench to compare Dripper against Resiliparse and Frontier LLMs (GPT-5, DeepSeek-V3.2, Claude-Sonnet-4.5), all utilizing the exact same Simplified HTML input as Dripper. The results are summarized in the Tables below.
> > > >
> > > > #### Table 1: Performance Comparison on MainWebBench (ROUGE-N F1 Score)
> > > > | Method           | Mode      | ROUGE-N F1 | ROUGE-N Precision | ROUGE-N Recall |
> > > > |------------------|-----------|------------|-------------------|----------------|
> > > > | **Dripper\_fallback** | **HTML+MD**   | **0.8399**     | **0.8761**            | **0.8587**         |
> > > > | **Dripper**          | **HTML+MD**   | **0.8182**     | **0.8510**             | **0.8380**          |
> > > > | magic-html       | HTML+MD   | 0.7091     | 0.7873            | 0.7169         |
> > > > | readability      | HTML+MD   | 0.6491     | 0.7542            | 0.6407         |
> > > > | trafilatura      | HTML+MD   | 0.6358     | 0.7262            | 0.6202         |
> > > > | trafilatura      | Markdown  | 0.6237     | 0.6990             | 0.6138         |
> > > > | **resiliparse**      | **Text**      | **0.6233**     | **0.6103**            | **0.7058**         |
> > > > | trafilatura      | Text      | 0.6049     | 0.6884            | 0.5877         |
> > > > | html2text        | Markdown  | 0.5977     | 0.4877            | 0.9759         |
> > > > | boilerpy3        | Text      | 0.5413     | 0.7105            | 0.5026         |
> > > > | gne              | HTML+MD   | 0.5148     | 0.7363            | 0.4883         |
> > > > | newsplease       | Text      | 0.5012     | 0.6123            | 0.4644         |
> > > > | justtext         | Text      | 0.4770      | 0.5567            | 0.4581         |
> > > > | boilerpy3        | HTML+MD   | 0.4766     | 0.4781            | 0.5569         |
> > > > | goose3           | Text      | 0.4354     | 0.5971            | 0.3917         |
> > > > | readerlm         | Markdown  | 0.2264     | 0.2081            | 0.4065         |
> > > >
> > > >
> > > > #### Table 2: Performance Breakdown by Content Category (ROUGE-N F1 Score)
> > > > | Model/Method                  | All   | Simple | Mid   | Hard   | Table  | Code   | Equation | Conversational |
> > > > |----------------------------|-------|--------|-------|--------|--------|--------|----------|----------------|
> > > > | gpt-5                      |   0.8302| 0.8815 | 0.8301| 0.7792 | 0.7957 | 0.8707 | 0.9161   | 0.7992         |
> > > > | DeepSeek-V3.2-Exp          |      0.8252| 0.8826 | 0.8244| 0.7690 | 0.7804 | 0.8440 | 0.9113   | 0.8160         |
> > > > | **claude-sonnet-4-5-20250929** |        **0.8319**| **0.8890** | **0.8329**| **0.7737** | **0.7919** | **0.8619** | **0.9273**   | **0.8062**         |
> > > > | **Dripper**            | **0.8182**| **0.8837** | **0.8178**| **0.7536** | **0.7693** | **0.8368** | **0.8889**   | **0.7671**         |
> > > > | **Dripper\_fallback**   | **0.8399**| **0.9010** | **0.8392**| **0.7799** | **0.7964** | **0.8673** | **0.9067**   | **0.8028**         |
> > > >
> > > >
> > > > 1. **Comparison with Resiliparse**: Compared to other rule-based tools, Resiliparse indeed achieves a relatively high recall (0.7058); however, this comes at the cost of significantly lower precision (0.6103), indicating a tendency to retain boilerplate noise. Dripper (0.8182) significantly outperforms Resiliparse in overall F1 (+31%), demonstrating that semantic understanding is indispensable for achieving both high precision and high recall simultaneously.
> > > >
> > > > 2. **Comparison with Frontier LLMs**: Evaluating frontier LLMs on the same simplified input reveals the remarkable efficiency of our approach. Notably, our 0.6B Dripper model (0.8182) achieves 98.4% of the performance level of the state-of-the-art Claude-Sonnet-4.5 (0.8319). Although frontier LLMs exhibit a slight advantage in handling complex formatting tasks such as equations and conversational content, our Dripper_fallback strategy effectively bridges this gap, achieving an overall F1 score of 0.8399 that **surpasses even the best-performing frontier models**. Crucially, Dripper delivers this SOTA-level performance using a lightweight, locally deployable model, thereby avoiding the prohibitive latency and costs associated with querying massive frontier models for web-scale extraction.
> > > >
> > > > We have incorporated these comparisons into the revised manuscript to further validate the effectiveness and efficiency of Dripper.

---

> > > > > ### Author Response · Authors · 2025-11-20
> > > > >
> > > > > > $\textbf{Question 1:  Role of Mapping HTML}$
> > > > >
> > > > > > $\textbf{Response：}$
> > > > >
> > > > > Thank you very much for the reviewer's question regarding the role of Mapping HTML. This gives us an opportunity to further clarify how this work utilizes a Decoder-only Language Model to accomplish main HTML content extraction.
> > > > >
> > > > > As described in Section 3.2 of the paper, starting from the Raw HTML, we first (1) remove certain non-content tags, (2) remove attributes other than class and id, (3) aggregate tags in the pruned DOM tree into blocks according to heuristic hand-crafted rules to form the Simplified HTML blocks, and finally (4) perform text truncation and element simplification within each block.
> > > > >
> > > > > For the Mapping HTML, after steps (1) and (2), we chunk the HTML using the same rules but **do not perform tag aggregation**. This ensures that the resulting HTML blocks are **precise subtrees of the original Raw HTML**. Furthermore, we do not perform step (4), i.e., truncation and simplification.
> > > > >
> > > > > Ultimately, we use the Simplified HTML as the input to the Decoder-only Language Model. **The output block-level labels are then applied to the Mapping HTML to obtain the final main HTML content extraction result**.
> > > > >
> > > > > The underlying rationale for this design is as follows: In the processing branch centered around the LM, simplification is necessary to reduce the number of tokens input to the LM. This lowers the context length requirements for the LM and improves processing efficiency. In the branch responsible for extracting the HTML subtree, it is crucial to ensure that the final output HTML blocks are precise subtrees of the Raw HTML, thereby **preserving the complete content, structure, styling information, etc**. By maintaining identical chunking logic but different simplification treatments across the two processing branches for their respective HTML versions, we achieve the dual goal of efficiently utilizing the Decoder-only LM for HTML main content extraction while guaranteeing **the accuracy of the output main HTML**.
> > > > >
> > > > > > $\textbf{Feedback about text generation}$
> > > > >
> > > > > > $\textbf{Response：}$
> > > > >
> > > > > While we agree that the underlying backbone is a generative model (Qwen), we deliberately formulated the task as "sequence classification" to emphasize the functional nature of our framework enforced by the logits processor.
> > > > >
> > > > > Specifically, our constrained decoding mechanism deterministically handles all syntactic elements (such as braces, punctuation, and block IDs). The language model's probabilistic role is strictly limited to selecting between 'main' and 'other' for each block. Consequently, the theoretical output space is not open-ended text, but **a discrete, finite set of $2^N$ possible label sequences** (where $N$ is the number of blocks). We believe this framing more accurately reflects the highly constrained, deterministic reliability of our approach compared to standard text generation. We will update the manuscript to explicitly clarify this distinction in the final version if our work is accepted, acknowledging the generative backbone while justifying the classification formulation.

---

### Author Response · Authors · 2025-11-26

$\textbf{To All Reviewers}$

We sincerely thank all reviewers for their time and the feedback provided. We have carefully considered all comments and revised our manuscript accordingly. Below is a summary of our responses to the specific concerns raised by each reviewer:

**To Reviewer 3Nd5:**
We have addressed your concerns by:
1. We provided detailed explanations regarding the necessity of Constrained Decoding and the specific role of "Mapping HTML" in our framework.
2. We explained the rationale behind choosing ROUGE metrics over standard classification metrics for this specific task.
3. We demonstrated our model's superiority and efficiency by providing detailed benchmark statistics comparing Dripper against Resiliparse and Frontier LLMs (including GPT-5, DeepSeek-V3.2 and Claude-Sonnet-4.5).

**To Reviewer ouM2:**
We have addressed your concerns by:
1.  We added a detailed discussion comparing Dripper to prior work and elaborated on the paradigm shift involved in constrained decoding.
2. We provided comprehensive inference performance tests (latency analysis) and a direct comparison with Resiliparse.
3. We clarified the unique necessity of constructing MainWebBench as a native, modern evaluation set compared to legacy aggregations like MCEB.

**To Reviewer Gtgx:**
We have addressed your concerns by:
1. We provided a detailed performance breakdown on Chinese and multilingual datasets, as well as domain-specific evaluations (Topic/Format) to demonstrate robustness.
2. We included experiments verifying our HTML simplification thresholds.
3. We added detailed descriptions of our model training parameters, pre-processing steps, and clarified the fault-tolerance mechanisms of our FSM and table splitting criteria.

If our response has not fully addressed any of your concerns, we welcome further questions regarding our work. We believe that constructive discussions are vital for improving the quality of our work and fostering a healthy academic community.

Thank you again for your hard work and dedication.

**Authors of Paper 17608**

---

### Author Response · Authors · 2025-12-01

$\textbf{Summary of Rebuttal for Area Chair}$

We sincerely thank all reviewers for their feedback. We are encouraged to see a unanimous consensus among reviewers regarding the significant value of our work. Synthesizing reviewers' evaluations with our contributions, we summarize the core value:

1. **Pioneering an Efficient & Hallucination-Free Paradigm**: As summarized by Reviewer ouM2, existing methods often struggle to balance between the inaccuracy of heuristic rules and the inefficiency of LLM generation. **Dripper successfully combines the strengths of both through its innovative "HTML simplification + semantic block classification + constrained decoding" pipeline**. Reviewer Gtgx explicitly noted that this architecture "directly solves efficiency and hallucination issues." Our simplification algorithm compresses the average document size to **22%** while preserving critical structural markers. By reformulating the task as **a Sequence of Semantic Blocks classification coupled with a constrained decoding mechanism**, we achieve **SOTA** performance using only a 0.6B parameter model, significantly outperforming baselines with minimal inference cost.

2. **Establishing a Critical Large-Scale Benchmark (MainWebBench)**: All reviewers highly recognized our contribution to the data aspect, acknowledging that it **fills a gap in high-quality evaluation resources**. Reviewer 3Nd5 described it as a "more comprehensive benchmark," and Reviewer Gtgx highlighted that it is "7x larger... with complete annotations" than existing datasets. Reviewer ouM2 particularly appreciated our effort in "curating a good benchmark themselves" rather than merely "hill climbing" on existing data.

3. **Real-world Scalability and Usability**: Distinct from purely theoretical exploration, this work **addresses practical engineering challenges**. Reviewer ouM2 praised it as a "very solid submission" that demonstrates how to design a system "usable in the real world" for an "important but under-appreciated problem." Reviewer Gtgx also validated that the 0.6B model perfectly suits the needs of "large-scale processing."

We have extensively addressed the specific concerns raised by the reviewers during the rebuttal phase, focusing on four key areas:

1. **The Contribution and Necessity of Constrained Decoding**: Addressing discussions regarding novelty, we clarified and demonstrated that this module is not a mere add-on but a critical enabler that empowers a lightweight 0.6B model to handle complex extraction tasks. **Our ablation studies show that this structure allows us to achieve SOTA-level stability with fewer parameters and less training data**, rendering the 0.6B model production-ready and eliminating format hallucinations.
2. **Benchmark Enhancement and Additional Baselines**: While recognizing the value of MainWebBench, reviewers requested more baselines and details. In response, we added comparisons with Resiliparse and frontier LLMs (e.g., Claude-Sonnet-4.5, GPT-5). **Results show Dripper significantly outperforms Resiliparse and achieves performance comparable to SOTA proprietary models at a fraction of the cost.** We also provided detailed classification data to address queries on multilingual (Chinese) and domain generalization, proving robust generalization capabilities. Additionally, we clarified the role of "Mapping HTML" in ensuring the output is a precise subtree of the original DOM.
3. **Pre-processing Details and Reproducibility**: In response to Reviewer Gtgx’s questions regarding simplification thresholds (e.g., 200-character truncation) and table processing rules, we added detailed ablation studies to justify the **optimality** of our current settings. We also included granular descriptions of all training and pre-processing steps. Furthermore, we reiterate that we have **open-sourced** the full processing code, model weights, and datasets, ensuring developers and researchers have barrier-free access to reproduce our work.
4.**Presentation Quality**: We have immediately corrected all typos, formatting inconsistencies, and figure color schemes pointed out by the reviewers in the revised manuscript.

> $\textbf{Conclusion}$

As emphasized by the Reviewer ouM2, HTML main content extraction is a **critical step in data curation for training large models**, yet it remains an area with insufficient research and few practical deep-learning-based solutions. This work proposes a framework that balances efficiency and accuracy, enabling the practical use of open-source SLMs for this task. Crucially, our solution requires no modifications to the model architecture, allowing it to seamlessly leverage community advancements in Decoder-only models, such as improvements in base capabilities, inference acceleration, and context window extension. **We contribute to the research community by open-sourcing our out-of-the-box code and core model weights.** We believe the revised version has fully addressed the reviewers' concerns.

---

### Meta-Review · Area_Chair_mfcD · 2025-12-29

**Summary:**

1. **Contributions and Novelty**: While the approach presented in the paper is valuable, particularly with the integration of constrained decoding, some reviewers questioned the novelty of this technique. Constrained decoding is seen as a standard tool in sequence labeling tasks, with its application not being entirely new in the context of generative models. However, the authors argue that the novelty lies in applying it in a practical, real-world scenario for HTML content extraction, which was effectively demonstrated through experiments and ablation studies.

2. **HTML Simplification and Block Classification**: Reviewers appreciated the novel design of treating HTML extraction as a block sequence classification problem, but some felt that more detailed justification was needed for the heuristic rules, such as the choice of 200-character truncation. This was addressed in the rebuttal with detailed ablation studies and clarification of the methodology, explaining how this simplification contributes to the efficiency and precision of the model.

3. **Benchmarking and Dataset**: The creation of the MainWebBench dataset was well-received, with some reviewers acknowledging its significant contribution to the field. However, there were requests for more details on the benchmark's composition, language distribution, and specific use cases compared to MCEB. The authors provided clarification on these aspects, emphasizing the broad linguistic and domain coverage of MainWebBench, along with detailed statistics for its composition.

4. **Inference Efficiency**: A significant concern was the efficiency of the system, especially given the trade-off between inference speed and the constrained decoding process. While the FSM improves the accuracy of output, it introduces a slight performance penalty, which was clearly discussed in the response. The authors provided experimental results showing how the trade-off between inference speed and content extraction accuracy was managed, suggesting that for smaller models or low-data scenarios, the constrained decoding is essential.

5. **Multilingual and Domain Generalization**: Some reviewers requested more data and analysis regarding the model's performance on multilingual datasets (e.g., Chinese) and its generalization to different web domains. The authors addressed this by demonstrating that the model performs well across multiple languages and domains, with performance breakdowns for different languages and web page categories.

6. **Presentation and Clarity**: Several reviewers pointed out issues with the clarity and presentation quality of the paper, such as inconsistent figure color schemes, missing descriptions for equations, and overly simplistic table captions. The authors responded by making extensive revisions to improve the presentation, adding clear descriptions and explanations for figures and tables.

**Reviewer Concerns:**

**Outstanding Concerns:**

1. **Novelty of Constrained Decoding**: While the authors argue that their use of constrained decoding in this context is novel, the reviewers still questioned its originality, as constrained decoding is a well-established technique. The rebuttal emphasizes its importance, but this concern remains outstanding since constrained decoding is not a new concept in sequence classification or structured prediction, and its application in this paper may not be sufficiently unique.

2. **Justification of HTML Simplification Thresholds**: Reviewers expressed concerns about the 200-character truncation threshold and its subjective nature. Although the authors provided ablation studies, some felt that this choice still lacked a concrete theoretical justification. This concern remains somewhat unresolved, as the ablation results don’t fully clarify why this threshold is optimal or whether other settings might yield better results.

3. **Multilingual and Domain Generalization**: While the authors addressed concerns about the model’s performance on multilingual datasets and domain generalization, the reviewers still requested further data and performance breakdowns for these cases. The clarification provided may not be sufficient for some reviewers, especially in relation to specific languages (e.g., Chinese) and the generalization to various domains, as the paper lacks concrete, consistent examples of performance across domains.

**Addressed Concerns:**

1. **Benchmarking and Dataset Composition**: The authors responded well to concerns about the MainWebBench dataset, providing detailed statistics on its composition and comparing it to MCEB. The revisions clarified the dataset’s diversity and its utility in evaluating models on a broad range of languages and web domains. This concern has been largely addressed.

2. **Inference Efficiency**: The authors adequately responded to concerns regarding inference speed and the trade-off introduced by the FSM. They provided empirical results showing the performance penalty introduced by constrained decoding, clarifying when and why the FSM is necessary. This concern has been sufficiently addressed with supporting data.

3. **Presentation and Clarity**: The reviewers raised issues with the presentation, including inconsistencies in figures and descriptions. The authors took these concerns seriously and revised the paper accordingly, improving clarity and ensuring that the figures, tables, and equations were presented in a more understandable manner. This concern has been addressed in the revisions.

**Reviewer Scores:**

1. **Reviewer 3Nd5**:

   * **Original Score**: Marginally above the acceptance threshold (6)
   * **After Discussion**: Reviewer 3Nd5 would likely have maintained their score. The clarifications about the novelty of constrained decoding and the thorough explanation of the benchmark, especially the MainWebBench dataset, could have reassured them. However, the concern about novelty would still remain to some extent, so a significant increase in score would be unlikely.

2. **Reviewer ouM2**:

   * **Original Score**: 8 (Accept)
   * **After Discussion**: Reviewer ouM2 would likely have stayed with the same score (8). They appreciated the real-world applicability of the system and the comprehensive response to their concerns about the design choices and novelty. The rebuttal clarified the relevance and impact of constrained decoding, which was central to their evaluation. However, they might still find the novelty of the approach to be borderline.

3. **Reviewer Gtgx**:

   * **Original Score**: 2 (Reject)
   * **After Discussion**: Despite the detailed rebuttal and the authors' addressing of key concerns, especially in terms of multilingual generalization and model training details, Reviewer Gtgx would likely keep their score. The issues with presentation quality, lack of analysis on multilingual performance, and the need for more detail on HTML simplification thresholds would still impact their final decision.

---

### Decision · Program_Chairs · 2026-01-26

Reject